# Nonlinear response of the Antarctic Ice Sheet to late-Quaternary sea level and climate forcing

Michelle Tigchelaar[1], Axel Timmermann[2,3], Tobias Friedrich[4], Malte Heinemann[5], and David Pollard[6]

[1]Center for Ocean Solutions, Stanford University, Palo Alto, CA, USA
[2]Center for Climate Physics, Institute for Basic Science, Busan, South Korea
[3]Pusan National University, Busan, South Korea
[4]Department of Oceanography, University of Hawaiʻi at Mānoa, Honolulu, HI, USA
[5]Institute of Geosciences, Kiel University, Kiel, Germany
[6]Earth and Environmental Systems Institute, Pennsylvania State University, University Park, PA, USA

*Correspondence to:* M. Tigchelaar (mtigch@stanford.edu)

**Abstract.** Antarctic ice volume has varied substantially during the late-Quaternary, with reconstructions suggesting a glacial ice sheet extending to the continental shelf break, and interglacial sea level highstands of several meters. Throughout this period, changes in the Antarctic Ice Sheet were driven by changes in atmospheric and oceanic conditions and global sea level, yet so far, modeling studies have not addressed which of these environmental forcings dominate, and how they interact in the dynamical ice sheet response. Here we force an Antarctic Ice Sheet model with global sea level reconstructions and transient, spatially explicit boundary conditions from a 408 ka climate model simulation, not only in concert with each other but, for the first time, also separately. We find that together, these forcings drive glacial-interglacial ice volume changes of 12-14 m SLE, in line with reconstructions and previous modeling studies. None of the individual drivers – atmospheric temperature and precipitation, ocean temperatures, sea level – single-handedly explains the full ice sheet response. In fact, the sum of the individual ice volume changes amounts to less than half of the full ice volume response, indicating the existence of strong nonlinearities and forcing synergy. Both sea level and atmospheric forcing are necessary to create full glacial ice sheet growth, whereas the contribution of ocean melt changes is found to be more a function of ice sheet geometry than climatic change. Our results highlight the importance of accurately representing the relative timing of forcings of past ice sheet simulations, and underscore the need for developing coupled climate-ice sheet modeling frameworks that properly capture key feedbacks.

## 1 Introduction

At a time when the future of the Antarctic Ice Sheet (AIS) is both critical and highly uncertain (Joughin and Alley, 2011; Church et al., 2013; DeConto and Pollard, 2016), exploring its past behavior can lend insight to its sensitivity to external forcing. Records show that during the late-Quaternary (roughly the past one million years), the AIS contributed to both glacial sea level drops of more than 10 m (RAISED Consortium et al., 2014), as well as rapid deglacial sea level rise (Carlson

and Clark, 2012) and interglacial sea level highstands of several meters (Dutton et al., 2015). Throughout this period, Antarctic mass balance changes were driven by a wide spectrum of external forcings: changes in atmospheric temperatures, accumulation rates, oceanic conditions, and sea level (Tigchelaar et al., 2018). There is substantial spatial variability in the sensitivity of the AIS to these forcings, as indicated for example by surface-exposure chronologies since the Last Glacial Maximum (LGM)

(e.g., RAISED Consortium et al., 2014; Hillenbrand et al., 2014; Spector et al., 2017; Goehring et al., 2019). So far however, the relative contributions of different external drivers of past AIS variability and their synergies have not been quantified in modeling studies. Here we address how different forcing agents interact during the last four glacial cycles using a set of experiments with an AIS model.

Unlike the Greenland Ice Sheet, the AIS has large marine-based margins. The ice shelves surrounding the AIS have a
buttressing effect, and therefore play an important role in determining its stability. Disintegration of ice shelves can lead to rapid discharge from and acceleration of the grounded ice sheet, in particular when the bed deepens towards the ice sheet interior (a process referred to as 'marine ice sheet instability') (Schoof, 2007; Joughin and Alley, 2011). The importance of the Antarctic marine margins for ice sheet stability means that both the marine and the atmospheric environment contribute to ice volume changes. The accelerated mass loss of Pine Island Glacier over the past few decades for instance, has been attributed
to enhanced sub-shelf melting in response to warming oceans and changing ocean circulation (Jacobs et al., 2011; Pritchard et al., 2012). The 2002 collapse of the Larsen B Ice Shelf on the other hand, is thought to be the result of preconditioning by a warming atmosphere (van den Broeke, 2005). During the glacial cycles of the late-Quaternary, changes in eustatic sea level (Fig. 1C) further impacted Antarctic ice shelves: changes in the ice flux at the grounding line turn grounded ice into floating ice during sea level rise, and floating ice into grounded ice during sea level drops (Schoof, 2007). Previous modeling
studies of late-Quaternary AIS evolution have identified global sea level as an important pacemaker, especially for the West Antarctic Ice Sheet (WAIS) (Ritz et al., 2001; Huybrechts, 2002; Pollard and DeConto, 2009). Finally, changes in temperature and circulation patterns drive changes in accumulation rates that can affect both the marine margins and interior ice sheet. Future projections of AIS evolution suggest that in a warming world, accumulation rates will increase as a result of increased atmospheric moisture content, particularly leading to growth of the East Antarctic Ice Sheet (EAIS) (Huybrechts et al., 2004;
Frieler et al., 2015; Medley and Thomas, 2019).

Late-Quaternary climate change is ultimately caused by variations in earth's axial tilt and orbit around the sun (Milankovitch, 1941), i.e., precession, eccentricity and obliquity (Fig. 1A,B). These lead to changes in incoming solar radiation that cause a global climate and carbon cycle response that make changes in atmospheric greenhouse gas concentrations – primarily $CO_2$ (Fig. 1C) – an additional driver of long-term climate variability (Shackleton, 2000). Different climate variables respond differ-
ently to each of these forcings, resulting in a rich spectrum of Southern Hemisphere climate variability in both reconstructions (e.g., Steig et al., 2000; Gersonde et al., 2005; Cortese et al., 2007; Jouzel et al., 2007; Ho et al., 2012) and simulations (e.g., Huybers and Denton, 2008; Menviel et al., 2008; Timmermann et al., 2009; He et al., 2013; Timmermann et al., 2014).

Up to this point, Antarctic modeling studies have not considered how these various forcings interact in driving ice volume changes. Previous studies have either used heavily parameterized climate forcing (Ritz et al., 2001; Huybrechts, 2002; Pollard
and DeConto, 2009) or simplified climate and ice sheet configurations (de Boer et al., 2013; Stap et al., 2014); have focused

on equilibrium simulations of specific time periods (Golledge et al., 2012); or applied indexed interpolations of extreme climate states (Maris et al., 2015). All of these studies assume that Southern Hemisphere climate variables vary in pace with either Antarctic temperature reconstructions (Petit et al., 1999) or the benthic oxygen isotope record (Lisiecki and Raymo, 2005). They thus ignore the spatial and temporal heterogeneity of late-Quaternary climate variability, and preclude a better understanding of how different drivers interact.

The aim of this study is to better understand the individual and combined roles of sea level and climate variability in driving AIS evolution during the late-Quaternary. To that end we have forced a state-of-the-art AIS model with spatially-varying and time-evolving atmospheric temperature, precipitation and ocean temperature fields from a climate model simulation over the last four glacial cycles, as well as changes in eustatic sea level from Northern Hemisphere ice sheets. This work builds on Tigchelaar et al. (2018), which used a similar modeling setup but did not isolate individual drivers. We conduct a number of sensitivity experiments to explore the separate role and synergy of individual forcings and mechanisms contributing to past ice sheet variability. Looking at individual forcings allows us to identify which are important, which need modeling improvement, and how they might interact nonlinearly in future Antarctic change. These simulations can also be used to aid in interpretation of the rich AIS deglaciation record (RAISED Consortium et al., 2014; Hillenbrand et al., 2014; Spector et al., 2017; Goehring et al., 2019).

Section 2 provides a detailed overview of our climate and ice sheet modeling setup. In Sect. 3 the main results are presented, with Sect. 3.1 discussing late-Quaternary climate variability, Sect. 3.2 and 3.3 describing the ice sheet response to all and individual drivers, and Sect. 3.4 discussing the responsible mechanisms. Section 4 summarizes our results and discusses their implications.

## 2 Methods

The late-Quaternary orbital and greenhouse gas forcing shown in Fig. 1 is used to drive a transient simulation with an Earth system model of intermediate complexity (EMIC) over the last four glacial cycles (Sect. 2.1). Climate anomalies from this simulation, together with time-varying global sea level (Fig. 1c), are then used as boundary conditions for various sensitivity experiments (Sect. 2.2.3) with the Penn State University ice sheet model (PSU-ISM; Sect. 2.2.1) according to the equations outlined in Sect. 2.2.2. Figure 2 shows a schematic illustration of this modeling setup.

### 2.1 Climate model

Our ice sheet model experiments are driven with transient climate anomalies spanning the last four glacial cycles (408 ka to present) (Timmermann et al., 2014; Friedrich et al., 2016; Timmermann and Friedrich, 2016), derived from a simulation with the EMIC LOVECLIM (Goosse et al., 2010), which consists of coupled atmospheric, ocean-sea ice and vegetation components. The atmospheric component of LOVECLIM, ECBILT, is a spectral T21 ($\sim$5.625$^\circ\times$5.625$^\circ$), three-level model based on the quasi-geostrophic equations, extended by estimates of the ageostrophic terms (Opsteegh et al., 1998). The model contains a

full hydrological cycle and includes physical parameterizations of diabatic processes (radiative fluxes, sensible and latent heat fluxes) in the thermodynamic equation.

CLIO, the ocean sea-ice component, is a $3° \times 3°$ primitive equation ocean general circulation model with twenty vertical levels, coupled to a thermodynamic-dynamic sea ice model (Goosse and Fichefet, 1999). It uses parameterizations to compute mixing along isopycnals, the effect of mesoscale eddies on diapycnal transport and downsloping currents at the bottom of continental shelves. Finally VECODE is a terrestrial vegetation model that consists of two plant functional types and non-vegetated desert zones (Brovkin et al., 1997). Each land grid cell is assumed to be partially covered by these three land cover types, based on annual mean temperature and rainfall amount and variability.

For the transient climate model simulation, LOVECLIM was forced with time-evolving orbital parameters (Berger, 1978) and reconstructed atmospheric greenhouse gas concentrations ($CO_2$, $CH_4$ and $N_2O$) (Lüthi et al., 2008). The corresponding orbital forcing, annual mean and seasonal insolation changes and $CO_2$ time series are shown in Fig. 1. In addition, Northern Hemisphere ice sheet conditions were obtained from a transient experiment conducted with the Climate and Biosphere Model, version 2 (CLIMBER-2), coupled to the Northern Hemisphere Simulation Code for Polythermal Ice Sheets (SICOPOLIS) ice sheet model (Ganopolski and Calov, 2011). Orography, albedo and ice mask variations from this simulation are interpolated onto the LOVECLIM grid, where in the presence of land ice, the grid point albedo is set to 0.7 and the vegetation mask is modified. The orography, albedo and ice mask of the AIS remain constant throughout the simulation. Similarly, time-evolving Antarctic melt water fluxes are not fed back into LOVECLIM. The implications of this lack of ice sheet-climate coupling will be explored in the Discussion.

The orbital, greenhouse gas and ice sheet conditions are applied with a boundary acceleration factor of five (Timm and Timmermann, 2007; Timmermann et al., 2014). The acceleration technique is based on the assumption of relatively fast equilibration of surface variables to slow external drivers; it thus mostly affects the representation of deep ocean currents (Timm and Timmermann, 2007), but not of surface and thermocline processes that matter for our experiments. This means that 200 model years correspond to 1000 calendar years. The LOVECLIM simulation is conducted using LGM ocean bathymetry (Roche et al., 2007) in order to avoid the internally generated Atlantic meridional overturning circulation oscillations described in Friedrich et al. (2010). While the climate model run closely follows the methodology of Timmermann et al. (2014), here the longwave radiative effect of $CO_2$ was amplified by a factor of 1.97, based on model-proxy comparisons using 63 globally-distributed SST-reconstructions (Friedrich et al., 2016). The resulting net climate sensitivity amounts to $\sim 4$ °C per $CO_2$ doubling (Timmermann and Friedrich, 2016) and yields a more realistic glacial-interglacial amplitude in surface temperatures compared to paleo-proxy data.

## 2.2 Ice sheet model

The 408 ka climate anomalies from LOVECLIM are used to force a number of sensitivity experiments with the PSU-ISM (Fig. 2; Pollard and DeConto, 2009, 2012a; DeConto and Pollard, 2016; Pollard et al., 2016). Previous simulations with this ice sheet model driven by parameterized climates produced a realistic LGM state and subsequent deglacial retreat (Mackintosh

et al., 2011; Briggs et al., 2013, 2014; Pollard et al., 2016, 2017), to within general levels of uncertainty within the paleo-data, and also the modern state of grounded and floating ice (Pollard and DeConto, 2012a, b; Pollard et al., 2016).

The model is based on a combination of the scaled shallow ice and shallow shelf approximations, and calculates ice velocity across the grounding line using an ice flux parameterization (Schoof, 2007). Basal sliding on unfrozen beds is calculated using a standard drag law, with the basal sliding coefficients derived from a simple inverse method (Pollard and DeConto, 2012b). Bedrock deformation is modeled as an elastic lithospheric plate above local isostatic relaxation; the initial and equilibrium bedrock topography and ice-load state is taken to be modern observed (Bedmap2; Fretwell et al., 2013). The model includes vertical diffusion of heat and storage in bedrock below the ice, which is heated from below by a uniform geothermal heat flux for the EAIS and WAIS (Pollard and DeConto, 2012a).

The model is discretized on a polar stereographic grid. Due to limited computational resources and the long timescale of the simulations we used a relatively coarse resolution of 40 km, though note that previous studies with this model have shown that results are quite independent of horizontal resolution (Pollard et al., 2015, Supplemental Information).

### 2.2.1 Present-day mass balance and climate forcing

Present-day surface climate forcing – specifically, annual mean atmospheric temperature $T_a^{obs}$ (Fig. 3a; van de Berg et al., 2006) and accumulation $P^{obs}$ (Fig. 3b; Comiso, 2000) – is obtained from the ALBMAP v1 database at 5 km resolution (Le Brocq et al., 2010) and interpolated onto the ice model grid. A lapse rate correction of $\gamma = 0.008\,°C\,m^{-1}$ is applied to the atmospheric temperature to correct for differences between observed ($z^{obs}$; Le Brocq et al., 2010) and model ($z$) surface elevation. The seasonal cycle in atmospheric temperature is parameterized as a sinusoidal cycle with a range of 20 °C at sea level, increasing linearly with elevation to 30 °C at 3000 m and above (Pollard and DeConto, 2012a), giving $T_a^{obs}(\tau)$. Surface melt rates are calculated using a positive degree-day (PDD) scheme (Reeh, 1991) that uses different coefficients for ice ($8\,kg\,m^{-2}\,°C^{-1}$) and snow ($3\,kg\,m^{-2}\,°C^{-1}$) and allows for seasonal refreezing as well as diurnal and synoptic variability (Pollard and DeConto, 2012a). Present-day accumulation rates in the model do not contain a seasonal cycle, but are split into rain and snow based on monthly temperatures. In the ocean, the model interpolates modern annual mean 400 m-depth ocean temperatures $T_o^{obs}$ from the World Ocean Atlas (Locarnini et al., 2010) onto the ice sheet model grid (Fig. 3c). In areas outside the range of the Locarnini et al. (2010) dataset, ocean temperatures are propagated underneath the ice shelves using a nearest neighbor interpolation.

An important component of AIS modeling is the treatment of ice shelf processes and the ice-ocean interface. While in reality melting at the ice shelf-ocean interface is a function of ocean temperature, salinity and circulation in the ice shelf cavity (Jacobs et al., 1992), most ice sheet models used for long-term simulations make use of parameterizations based on sub-surface ocean temperatures alone. This ice model follows the parameterization developed by Martin et al. (2011) for the PISM-PIK model, where oceanic melt is a function of the difference between ocean temperature and the depth-varying freezing temperature of ocean water, with a quadratic dependency on this temperature difference (Holland et al., 2008; Pollard and DeConto, 2012a):

$$\text{OMB} = \frac{K K_T \rho_w c_w}{\rho_i L_f} \left| T_o - T_f \right| (T_o - T_f) \tag{1}$$

Here $K_T$ is the transfer coefficient for sub-ice oceanic melting (15.77 m a$^{-1}$ K$^{-1}$), $\rho_w$ is the ocean water density (1028 kg m$^{-3}$), $\rho_i$ is the ice density (910 kg m$^{-3}$), $c_w$ is the specific heat of ocean water (4218 J kg$^{-1}$ K$^{-1}$), and $L_f$ is the latent heat of fusion (0.335×10$^6$ J kg$^{-1}$). $T_o$ is the specified ocean temperature, and $T_f$ is the ocean freezing point at ice-base depth, assuming a salinity of 34.5 psu. The salinity – chosen to represent values at typical depths of Circumpolar Deep Water, not considering ice melt – has a very small effect on basal melt rates and is therefore kept constant. $K$ is an additional tuning parameter which – as in Pollard et al. (2015, Supplemental Information) – is set to be spatially uniform ($K$=3). This yields reasonable patterns of modern sub-ice shelf melt (ranging from near-zero to ~8.5m a$^{-1}$ (Pollard et al., 2017, Fig. S1a)) that are generally within ongoing estimates of empirical uncertainties and rapidly changing decadal trends (Depoorter et al., 2013; Rignot et al., 2013). Melt rates at vertical ice faces in direct contact with the ocean are calculated by multiplying the area of each vertical face with the oceanic melt rates at that grid point.

Calving rates at the ice shelf edge are parameterized based on the large-scale stress field, represented by the horizontal divergence of the ice shelf (Pollard and DeConto, 2012a; Nick et al., 2013). In recent years a new set of parameterizations was introduced to the ice sheet model representing sub-grid scale processes that have been hypothesized to significantly increase the sensitivity of the AIS to climatic forcing (Pollard et al., 2015). These parameterizations include increased calving due to hydrofracturing by surface melt and rainfall draining into crevasses (Nick et al., 2013), as well as structural failure at the grounding line when the vertical face of ice cliffs is too tall ('cliff failure') (Bassis and Walker, 2012). Combined, these two mechanisms have the potential to significantly reduce ice shelf extent and buttressing in warm climates (DeConto and Pollard, 2016; Bell et al., 2018). It is worth noting that these parameterizations – whose validity continues to be debated (Edwards et al., 2019) – do not get triggered by our simulated late-Quaternary climate anomalies (Sect. 3.1).

The mass balance terms in this study are calculated from a file written at run time that stores accumulation (snow+rain), ablation (abl), oceanic melt (ocn), melting at vertical ocean faces (face) and calving (calv), averaged over the entire ice sheet area. The ablation term (abl) here represents the combined contributions of evaporation at the surface, melting at the base of the grounded ice sheet, and percolation of rain, surface melt water and frictional melt water to the base of the ice sheet, minus refreezing in the ice column. Evaporation and basal melting of grounded ice are both very minor, and surface melt dominates the percolation term; therefore we refer to the ablation term below as 'surface melt'.

### 2.2.2   Climate and sea level forcing over the last 408 ka

Instead of parameterizing the paleo-climate forcing of the late-Quaternary, as done in previous studies, we force the PSU-ISM with climate anomalies from the 408 ka transient experiment described in Sect. 2.1 (Tigchelaar et al., 2018). The climate forcing in the ice sheet model is updated every 1000 calendar years. Climate anomalies are calculated with respect to the LOVECLIM climatology over the last 200 model years (representing 1000 calendar years) and are bilinearly interpolated to the ice sheet model grid. The atmospheric temperature $T_a$ is modified by a lapse rate correction of $\gamma = 0.008$ °C m$^{-1}$ to account for surface elevation differences between the reference ice sheet geometry ($z^{\text{obs}}$; Le Brocq et al., 2010) and the simulated elevation at time

t ($z(t)$). Subsequently, monthly temperature anomalies are added to the present-day temperature field (Fig. 3a; Sect. 2.2.1):

$$T_a(t,\tau) = T_a^{obs}(\tau) - \gamma\left[z(t) - z^{obs}\right] + \left[T_a^{LC}(t,\tau) - T_a^{LC}(0,\tau)\right],\tag{2}$$

where $t$ indicates time in years, $\tau$ represents month of year, $\gamma$ is the lapse rate and superscripts 'obs' and 'LC' indicate present-day and LOVECLIM variables respectively.

Because the ice sheet model does not include a seasonal cycle for present-day precipitation, precipitation anomalies are calculated with respect to annual mean precipitation. Instead of adding the anomalies to the present-day field, as done for atmospheric temperature, present-day precipitation ($P^{obs}$) is multiplied with the ratio of monthly LOVECLIM precipitation at time $t$ ($P^{LC}(t,\tau)$) to present-day LOVECLIM precipitation ($P^{LC}(0)$):

$$P(t,\tau) = P^{obs}\left[\frac{P^{LC}(t,\tau)}{P^{LC}(0)}\right].\tag{3}$$

This is done to ensure that precipitation rates do not go below zero. Annual mean ocean temperature anomalies from the 400 m depth level in LOVECLIM are added to the ice model field as

$$T_o(t) = T_o^{obs} + \left[T_o^{LC}(t) - T_o^{LC}(0)\right].\tag{4}$$

The ocean temperature is set not to decrease below -2.18 °C, which is the freezing temperature of sea water with a salinity of 34.5 psu at 400 m depth (Beckmann and Goosse, 2003).

Figures 3d-f show the differences between LOVECLIM and observed present-day climate. Modeled atmospheric temperatures over the Antarctic interior are too high, even when corrected for differences in observed surface elevation and the T21 spectral representation of Antarctic orography in LOVECLIM (Fig. 3d). Present-day Antarctic precipitation is characterized by a temperature-driven low accumulation regime (<50 $\text{mm}\,\text{a}^{-1}$) over the Antarctic interior and much higher precipitation rates in coastal areas (>1000 $\text{mm}\,\text{a}^{-1}$) as a result of cyclonic activity and topographic uplift (Bromwich, 1988). LOVECLIM does

not capture the complex coastal topography of Antarctica well, and therefore underestimates coastal precipitation, distributing it over the ice sheet interior instead (Fig. 3e; Maris et al., 2012). Sub-surface ocean temperatures in LOVECLIM are generally too low in the Southern Ocean, except below the shelves, where they are higher than in the World Ocean Atlas climatology. The lower LOVECLIM temperatures might be related to the fact that for present-day climate, minimum sea ice extent is overestimated (Roche et al., 2012). It should also be noted however that the observed climatology in the Southern Ocean is

based on a relatively low number of observations, especially close to the Antarctic continent (Locarnini et al., 2010). In any case, LOVECLIM climate anomalies rather than the full fields are applied to the ice sheet model to avoid the propagation of LOVECLIM biases into the ice sheet evolution. As will be discussed in Sect. 3.1, in spite of present-day biases, LOVECLIM generally simulates the late-Quaternary climate evolution well.

In addition to climate anomalies, the ice sheet model is forced with time-evolving eustatic sea level. Sea level variations

are derived from Spratt and Lisiecki (2016) and are plotted in Fig. 1c. While the climate fields are updated every 1000 years, sea level evolves continuously. The PSU-ISM uses a standard Elastic Lithosphere Relaxing Asthenosphere model for bed depression and rebound under the varying ice load, and therefore does not include deformational, gravitational, and rotational

contributions to local sea level change. Such contributions would potentially act as negative feedbacks for ice retreat, and cause spatial variability between the East and West Antarctic Ice Sheets (Gomez et al., 2015). However, previous work that includes full-Earth coupling suggests this would likely only have small effects on our timescales (Gomez et al., 2013, 2015; Pollard et al., 2017), and it is currently not computationally feasible to run a full-Earth model for our 400 kyr simulations (though work is in progress to improve this, e.g., Gomez et al., 2018).

### 2.2.3 Sensitivity experiments

The main ice sheet model simulation is run for 408 ka and includes all drivers described in Sect. 2.2.2 (experiment 'all'). In order to isolate the effects of these individual external forcings on AIS variability and their interaction, we performed a series of sensitivity experiments that include only one or multiple drivers. The individual drivers are either the atmospheric forcing described by Eqs. (2) and (3), the ocean temperature forcing of Eq. (4) or the sea level variations from Spratt and Lisiecki (2016) (experiments 'atm', 'ocn' and 'sl', respectively). In addition to these singular forcing experiments, the model is forced with combinations of two of these three forcings (experiments 'sl+atm', 'sl+ocn' and 'atm+ocn'). These experiments are designed to quantify the synergistic response of the AIS to a variety of acting forcings. All sensitivity experiments are summarized in Table 1.

## 3 Results

### 3.1 Late-Quaternary climate forcing

The spatial and temporal evolution of atmospheric temperature, precipitation and sub-surface ocean temperatures are characterized by the first principal component (PC1) and the corresponding spatial pattern (EOF1) as shown in Fig. 3g-l. The full amplitude of this first mode at each point in space can be derived by multiplying the EOF1 map with the PC1 time series. As can be seen in Fig. 3j, annual mean surface temperatures over Antarctica are predominantly paced by changes in atmospheric $CO_2$ (Fig. 1c). Timmermann et al. (2014) showed that obliquity also contributes to annual mean temperature changes, by affecting annual mean insolation (Fig. 1b) and modulating the strength of the Southern Hemisphere westerlies. The dominant pattern of annual mean temperature changes is homogeneous, with a glacial-interglacial amplitude of $\sim$4-8 °C (Fig. 3g). We compare this to a composite of temperature reconstructions, calculated as the average of available long-term temperature records for each time in the past (Parrenin et al., 2013). The temporal evolution of simulated Antarctic air temperature is very similar, but the amplitude is underestimated by a factor of 1.5-2. This could partially be due to the fact that the LOVECLIM simulation does not include the lapse rate response to the evolving ice sheet height, but also points at an underestimation of polar temperature change in the climate model, especially during interglacials (Tigchelaar et al., 2018). In addition to annual mean temperatures, surface ablation rates are sensitive to changes in seasonal insolation (Huybers and Denton, 2008; Huybers, 2009; Tigchelaar et al., 2018), which is precessionally driven and shown in Fig. 1b.

Precipitation changes display a temporal evolution very similar to that of the atmospheric temperature PC1 (Fig. 3k), confirming that temperature is the dominant driver of precipitation over Antarctica. When compared to a composite of ice core accumulation reconstructions (Steig et al., 2000; Bazin et al., 2013; Vallelonga et al., 2013), LOVECLIM is shown to overestimate precipitation rates during early glacial times. Steig et al. (2000) describe how, when the AIS is expanding, the coastal ice core locations switch from a cyclonic-driven precipitation regime to one driven by temperature with increasing distance from the ice edge. The local precipitation evolution captured by the ice cores thus differs from the large-scale evolution captured by the principal component analysis. This ice sheet-climate feedback is not included in our LOVECLIM simulations.

The temporal evolution of sub-surface ocean temperatures in LOVECLIM (Fig. 3l) is similar to that of surface (not shown) and atmospheric temperatures (Fig. 3j). To our knowledge no reconstructions of intermediate water temperatures in the Southern Ocean exist, so we compare against a long sea surface temperature (SST) record from 54 °S (Ho et al., 2012) and a deep-sea temperature record from 41 °S (Elderfield et al., 2012). The LOVECLIM-simulated glacial-interglacial SST anomaly at the Ho et al. (2012) core location is about 8 °C (not shown), which is similar to the reconstructed amplitude of temperature variability – though the Last Interglacial warming is less pronounced in the climate model. At depth however, the simulated glacial-interglacial amplitude is about three times smaller than the Elderfield et al. (2012) reconstruction (not shown). The LOVECLIM-simulated ocean temperature anomalies close to the Antarctic continent are also small: zonal mean glacial-interglacial temperature anomalies at 65 °S and 400 m depth are about 0.6 °C, smaller than in some other models (Lowry et al., 2019) and with minimal interglacial warming (Tigchelaar et al., 2018). This means the magnitude of ocean forcing in our simulations is much smaller than thresholds for interglacial ice sheet collapse found in previous sensitivity studies (e.g. Sutter et al., 2016; Tigchelaar et al., 2018). The implications of this possible underestimation of ocean forcing on ice sheet evolution will be discussed further below.

## 3.2 Ice volume response to external forcing

Figure 4 shows the simulated response of Antarctic total, grounded, and floating ice volume to the individual and combined late-Quaternary forcings over the last four glacial cycles. With all forcings combined ('all'), the glacial-interglacial difference in ice volume is $\sim 8 \times 10^6$ km$^3$, or 12-14 m sea level equivalent (SLE) depending on the glacial stage (Fig. 4). During glacial periods, floating ice volume is reduced by about half the present day value ($\sim 7 \times 10^5$ km$^3$) (Fig. 4c). In our simulations, previous interglacials only contribute 1-2 m to global sea level ($\sim 1 \times 10^6$ km$^3$), with the deepest interglacial occurring at 210 ka (Termination IIIa). Tigchelaar et al. (2018) showed that local changes in summer insolation play an important role in amplifying interglacial ice loss.

The dominant spatial pattern of ice sheet thickness variability in the 'all' simulation, along with minimum (210 ka) and maximum (18 ka) grounding line extent, are shown in Fig. 5a. At its maximum extent, the grounding line reaches to the continental shelf break everywhere. The simulated minimum grounding line extent over the last 408 ka is very similar to present-day, with further retreat mostly of the Ross and Weddell ice shelves in West Antarctica, and the West and Shackleton ice shelves in East Antarctica. Changes in ice sheet thickness are most pronounced in those regions where the grounded ice

sheet expands, in particular the Ross and Weddell sectors, Amundsen Sea and Amery shelf. In the interior of the AIS, thickness changes are generally smaller, but mostly of the same sign.

Generally speaking, our complete-forcing simulation captures the main features of the Antarctic LGM and subsequent deglaciation well, to within the general level of uncertainties within the paleo-data (Pollard and DeConto, 2012a, b; Pollard et al., 2016; Tigchelaar et al., 2018). The simulated LGM grounding line position is in close agreement with reconstructions, as is the sequencing of regional deglaciation (Bellingshausen, followed by Amundsen, followed by Weddell, Ross and Amery) (RAISED Consortium et al., 2014). However, in our simulation retreat in the Ross sector occurs at least ∼2 ka earlier than reconstructions suggest, and there is an 'overshoot' of Siple Coast grounding lines at ∼6-4 ka. Addressing these discrepancies is the focus of ongoing work, including the large-ensemble simulations of e.g., Briggs and Tarasov (2013); Briggs et al. (2014); Pollard et al. (2017).

### 3.3 Nonlinear response to climate and sea level forcing

Not one of the individual drivers of late-Quaternary AIS variability – sea level, atmospheric temperature and precipitation, ocean temperatures – single-handedly explains the full ice volume evolution (Fig. 4a). Moreover, all of the individual forcings combined only account for less than half of the total ice volume changes, suggesting that they do not add linearly. The largest contribution in terms of both total and grounded ice volume comes from the atmospheric forcing, which explains about a third of glacial ice volume gain, and the entirety of interglacial ice volume loss (Fig. 4a,b). The case is different for floating ice: here sea level changes are responsible for most of the variability, as a lowering sea level converts floating into grounded ice (Fig. 4c; Schoof, 2007). Interestingly, for the floating ice volume, the sum of the individual simulations is not only smaller than, but also often not of the same sign as the floating ice volume changes in the 'all' simulation. When the ice sheet model is forced with two out of three forcings, sea level and atmospheric forcing together almost entirely explain the changes in both grounded and floating ice volume (Fig. 6).

Figure 5 shows where the individual drivers have the largest effect on the AIS. The sea level forcing drives expansion of the grounding line in the Amundsen, Ross, and Weddell Sea sectors, with small corresponding elevation changes (Fig. 5b). Atmospheric cooling leads to grounding line expansion primarily in the Amundsen, Weddell, and Amery regions, and also leads to thickening of most of the ice shelves (Fig. 5c). During interglacials, the atmospheric forcing causes retreat primarily of the West and Shackleton ice shelves. Due to the small magnitude of the simulated ocean temperature change, the oceanic forcing alone affects Antarctic ice volume only minimally. In fact, the dominant spatial pattern of ice thickness variability for the 'ocn' simulation only explains ∼10% of the variance, and is not driven by external forcing, but rather displays internally generated ice sheet variability in the Siple Dome region (Fig. 5d) with a period of ∼10 ka (not shown).

Combinations of external forcings lead to a more pronounced grounding line advance during glacials than in simulations with one single forcing (Fig. 5e-g). As noted above, sea level and atmospheric forcing combined (Fig. 5e) explain most of the grounding line and elevation changes simulated in the full run (Fig. 5a). A combination of sea level and ocean forcing (Fig. 5f) leads to grounding line expansion and ice sheet growth in the Weddell Sea sector, while atmospheric and ocean forcing combined (Fig. 5g) mostly cause ice sheet growth in the Ross Sea.

### 3.4 Mechanisms explaining ice volume changes

#### 3.4.1 Sea level forcing

Figure 7 depicts the response of grounded ice volume to the respective forcing in the different sensitivity runs, with corresponding mass balance changes shown in Fig. 8. As noted before, the impact of sea level forcing in isolation is to convert grounded into floating ice during periods of sea level rise, and the other way around during sea level drops (Fig. 7a). Changes in mass balance rates are a feedback to these changes in ice sheet configuration. Ice-sheet integrated surface melt rates (Fig. 8b) increase during glacial periods of sea level drop because the edges of the ice sheet – where all surface melt occurs – are lower in elevation, with associated higher temperatures. Calving rates (Fig. 8d) similarly increase during periods of low sea level, because the grounding line is positioned more equatorward (Fig. 5), increasing ice shelf divergence (Tigchelaar et al., 2018). On the other hand, ice-sheet integrated oceanic melt rates (Fig. 8c) decrease when sea level drops because the ice-ocean interface area is reduced. These mass balance feedbacks mostly cancel out in the net mass balance (Fig. 8e), explaining why total ice volume changes under isolated sea level forcing are small (Fig. 4).

#### 3.4.2 Atmospheric forcing

When atmospheric forcing is applied in isolation, grounded ice volume increases with decreasing surface temperature, whereas floating ice volume plateaus for ice-sheet averaged temperatures lower than ∼-34°C (Fig. 7b). In this case the mass balance response (Fig. 8) is a combination of both forcing and feedback. Surface melt rates (Fig. 8b) most directly follow the climatic forcing. As detailed in Tigchelaar et al. (2018), periods of high $CO_2$ and high summer insolation (Fig. 1) are marked by peaks in summer melt rates that also drive increases in calving rates (Fig. 8d). During these periods, the AIS retreats to areas that have lower accumulation rates (Fig. 3b), amplifying the forcing. In cold periods, a reduction in surface melt and calving leads to a small outward expansion of the grounding line (Fig. 5c), which causes the floating ice shelves to sit in climatologically warmer waters (Fig. 3c), increasing glacial ocean melt rates (Fig. 8c). The changes in ocean melt rates almost but not entirely balance the surface melt and calving rate changes, making the mass balance slightly positive during glacial periods (Fig. 8e).

#### 3.4.3 Ocean temperature forcing

Out of the three individual drivers, the ocean temperature forcing by itself leads to the least change in grounded and floating AIS volume, as seen in Figs. 4 and 5d. LOVECLIM-modeled ocean temperature anomalies are small (Fig. 3i), and lead to minor increases in grounded and floating ice thickness during glacial periods (Fig. 7c). The accompanying mass balance changes are similarly small (Fig. 8). Glacial expansion of floating ice area (Fig. 4c) brings ice shelves into areas with climatologically higher precipitation rates (Fig. 3b), leading to higher glacial accumulation rates (Fig. 8a). The reduced oceanic melt and increased accumulation are balanced by higher calving rates (Fig. 8d). It is important to note here that this small response to ocean temperature forcing is more likely a function of the low amplitude of the LOVECLIM-simulated ocean temperature forcing than it is indicative of low sensitivity of the AIS to changing ocean conditions, as will be discussed further below.

### 3.4.4 Combined forcings

Our sensitivity runs show that the simulated response of the AIS to late-Quaternary external drivers is a nonlinear superposition of a) a direct mass balance response to climate variations, b) sea-level induced conversion between grounded and floating ice, and c) areal expansion or contraction against climatological gradients. As shown in Fig. 7d-f, when all forcings combine, sea level is the dominant pace maker of both grounded and floating ice volume. Because sea level and atmospheric temperature vary in concert throughout the late-Quaternary (Fig. 1c, Fig. 3j), grounded ice volume also increases with lowering temperatures, while floating ice volume now decouples from atmospheric temperatures (Fig. 7e). The joint sea level and atmospheric forcing amplify each other in the total ice volume response, because the combination of shelf-to-sheet conversion (Fig. 4c) and reduced calving rates (Fig. 8d) allow the grounding line to migrate equatorward during glacial times (Fig. 5). This increases the ice sheet area – and thus the ice-sheet integrated accumulation rate (Fig. 8a) – leading to a net positive mass balance (Fig. 8e), and ice sheet growth.

With all forcings combined, the simulated ice sheet response is completely decoupled from the oceanic temperature forcing (Fig. 4f). During glacial periods, the grounding line is closer to warmer Circumpolar Deep Water (Fig. 3c), so that periods of high total ice volume are associated with high ocean temperatures beneath the ice shelves. This is also seen in Fig. 8c, where ice-sheet averaged oceanic melt rates in the 'all' simulation more closely follow those of the 'atm' run than the 'ocn' run. The spatial gradients in ocean temperature are thus larger and more important than temporal (glacial-interglacial) temperature variations (Fig. 3). The main exception to this is the Ross sector, where decreasing ocean temperatures allow for further ice expansion and grounding line migration during glacial times (compare e.g., Fig. 5c & g). Kusahara et al. (2015) also found oceanic melt rates to increase during the Last Glacial Maximum in response to grounding line migration, lending support to these findings. However, as shown in Tigchelaar et al. (2018), the low sensitivity of the modeled AIS to interglacial ocean conditions is likely a result of the low amplitude and resolution of the LOVECLIM ocean temperature forcing and lack of ice-ocean feedbacks in our modeling setup.

## 4 Discussion & Conclusions

Here we presented results from simulations of AIS evolution over the past 408 ka. In contrast to previous work which primarily used parameterized forcing, climate anomalies (atmospheric temperature, precipitation and sub-surface ocean temperatures) were directly derived from a transient simulation with the EMIC LOVECLIM. The simulated AIS has a glacial-interglacial amplitude of 12–14 m SLE, with the glacial grounding line extending almost entirely to the continental shelf break, and past interglacials showing limited retreat of 1–2 m SLE. Sensitivity experiments where atmospheric, oceanic and sea level forcing were applied in isolation or in pairs, showed that the combined effect of individual forcings is strongly nonlinear. Each of the individual forcings explains less than a third of the full response, and the sum of the individual forcing simulations is less than half of the glacial-interglacial amplitude with all forcings applied jointly. In our simulations, sea level and atmospheric forcing together explain most of the full response, both in terms of amplitude and pacing.

Our finding that ocean temperature forcing plays a limited role in driving changes in Antarctic ice volume contrasts with previous modeling studies of past and future AIS evolution (Golledge et al., 2015; DeConto and Pollard, 2016; Sutter et al., 2016), as well as observations of sustained sub-shelf ice loss in response to ongoing ocean warming at e.g., Pine Island Glacier (Jacobs et al., 2011; Pritchard et al., 2012). This is not surprising, given that the LOVECLIM-simulated ocean temperature anomalies are small (Figs. 3i,l), and ice sheet models typically need ocean warming of 2-5 °C to initiate interglacial WAIS collapse (Pollard and DeConto, 2009; DeConto and Pollard, 2016; Sutter et al., 2016; Tigchelaar et al., 2018). In the absence of paleo-reconstructions of near-Antarctic sub-surface ocean temperatures it is difficult to assess how realistic our LOVECLIM simulation is, though critical processes such as Antarctic Bottom Water formation are known to be poorly represented in low-resolution climate models (e.g., Snow et al., 2015), and previous studies have found LOVECLIM in particular to have more muted late-Quaternary temperature variability than other models (Lowry et al., 2019).

In addition, many regional oceanographic processes can affect the circum-Antarctic ocean environment beyond large-scale climate forcing. For example, the blocking effects of sea ice formation (Hellmer et al., 2012), the role of winds in pushing warm waters onto the continental shelf (Thoma et al., 2008; Steig et al., 2012), and the complex geometry of ice shelf cavities (Jacobs et al., 2011; De Rydt et al., 2014) have all been found to be important in observational and modeling studies of current and future oceanic melting of the WAIS ice shelves (Joughin et al., 2014). For that reason, using 400 m-depth Southern Ocean temperatures as the sole driver for sub-shelf melt may miss important near-Antarctic dynamics. Furthermore, melt water fluxes from the AIS have been found to lead to cooling of surface waters and warming at intermediate depth (Menviel et al., 2010; Weber et al., 2014), a feedback mechanism that could increase ice sheet loss (Golledge et al., 2014, 2019).

These processes can only really be captured in fully coupled ocean-atmosphere-ice sheet simulations at high resolution, something that is currently not feasible for the long timescales of late-Quaternary climate evolution. However, it should be possible to run shorter simulations – of e.g., the Last Interglacial or Marine Isotope Stage 11 – using such a setup, and perform a similar set of sensitivity experiments as done here. This would likely reveal additional nonlinearities as ice sheet and forcing are allowed to evolve together. The accumulation forcing for instance is similarly impacted by low climate model resolution and lack of ice-climate feedbacks. Time-evolving changes in orography and albedo can substantially alter atmospheric circulation patterns and associated rainfall (Steig et al., 2000; Maris et al., 2014; Steig et al., 2015).

The strongly nonlinear response of the AIS to different external forcing agents underscores the importance of driving the ice sheet model with accurately dated sea level and climate forcing. Previous modeling studies of past AIS evolution (Ritz et al., 2001; Huybrechts, 2002; Pollard and DeConto, 2009) have mostly bypassed this issue by assuming that both the sea level and climate forcing vary in concert with either Antarctic temperature reconstructions or the benthic $\delta^{18}O$ record. However, global sea level and global climate (i.e., $CO_2$) do not always vary in phase (Fig. 1c), and local climate conditions (e.g., through local insolation changes) can deviate substantially from global climate variability (Tigchelaar et al., 2018). At the same time, there are significant uncertainties in the timescales of especially Antarctic climate and $CO_2$ reconstructions (Lüthi et al., 2008; Bazin et al., 2013). Repeating the LOVECLIM climate simulations with a proper uncertainty range in atmospheric greenhouse gas concentrations would be computationally unfeasible, though future sensitivity runs with the ice sheet model could include artificial shifts in the phase relationship between the sea level and climate forcing to explore associated nonlinearities.

In our simulations, the sea level and climate forcing amplify each other during glacial AIS growth, while interglacial ice volume loss is almost exclusively driven by climate forcing (Fig. 4). Tigchelaar et al. (2018) showed that maximum interglacial ice loss occurs when high $CO_2$ concentrations coincide with high Southern Hemisphere summer insolation (Fig. 1). These precessionally-forced periods of warm summers are typically out of phase with eustatic sea level forcing, which is predominantly paced by warm Northern Hemisphere summers (Raymo et al., 2006). Our simulations therefore do not fully explore the response of the AIS to combined climate warming and rising sea levels, as they would co-occur in future climate change. So far, most modeling studies of future Antarctic ice sheet evolution (e.g., Joughin and Alley, 2011; Scambos et al., 2017; DeConto and Pollard, 2016) have not included changes in eustatic sea level. Further research should therefore explore whether – given the current configuration of grounding line and bedrock (Joughin and Alley, 2011; Joughin et al., 2014) – rising sea levels as a result of global warming would further increase or stabilize ice loss. Such future studies should make sure to include the deformational and gravitational components of future sea level rise through coupling with a full-Earth model (Gomez et al., 2018).

In response to changes in atmospheric and oceanic conditions and global sea level, Antarctic ice volume has varied by tens of $\mathrm{m\,SLE}$ throughout the late-Quaternary, and is expected to decrease in the future. In contrast to previous modeling studies, here we focused on the interaction of different external forcings driving Antarctic ice volume changes. Our sensitivity experiments with an Antarctic Ice Sheet model over the last four glacial cycles showed that the glacial-interglacial ice sheet response to environmental forcing is strongly nonlinear. Both atmospheric cooling and a transformation of dynamic regime by lowering sea level were found necessary to generate full glacial ice sheet growth. Our modeling setup likely underestimates the role of oceanic forcing, which remains largely unbound by the geologic record and needs to be further explored in a coupled climate-ice sheet modeling framework that can account for critical circum-Antarctic oceanographic processes.

*Author contributions.* MT, AT, and DP designed the ice sheet model experiments. TF ran the climate model simulations. DP developed the ice sheet model, and MH developed the ice sheet-climate coupling. MT ran the ice sheet model and analyzed the results; MT, AT, and DP contributed to the interpretation of the results. MT wrote the first draft of the paper; AT, TF, MH, and DP contributed substantially to its final version.

*Data availability.* Our ice sheet simulations are publicly available at https://climatedata.ibs.re.kr/grav/data/psu-love/.

*Competing interests.* The authors declare that they have no conflicts of interest.

*Acknowledgements.* This study was supported by National Science Foundation grant #1341311 and the Institute for Basic Science under grant IBS-R028-D1. The authors kindly thank Andrey Ganopolski for making available his 800 ka Northern Hemisphere ice sheet simulation

results, Barbara Stenni for sharing the full 314 ka TALDICE oxygen isotope record, Elke Zeller for her help with data archiving, and the University of Washington g-lunch for insightful discussions.

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

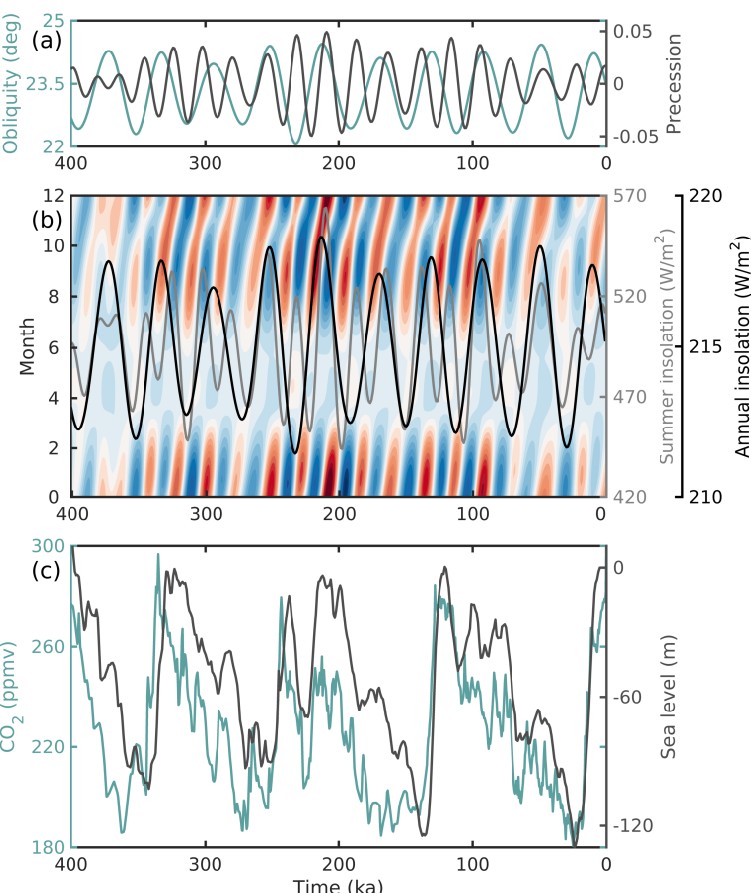

**Figure 1.** Climate drivers over the last 400 ka – (a) precession (grey) and obliquity (teal) (Laskar et al., 2004); (b) monthly insolation anomalies (colors, contours ranging from $\pm 65\,\mathrm{W\,m^{-2}}$), annual mean insolation (black) and summer insolation (grey) at 65 °S as a result of the orbital forcing in (a) (Laskar et al., 2004); (c) atmospheric $CO_2$ concentration (teal; Lüthi et al., 2008) and global sea level (m) (grey; Spratt and Lisiecki, 2016).

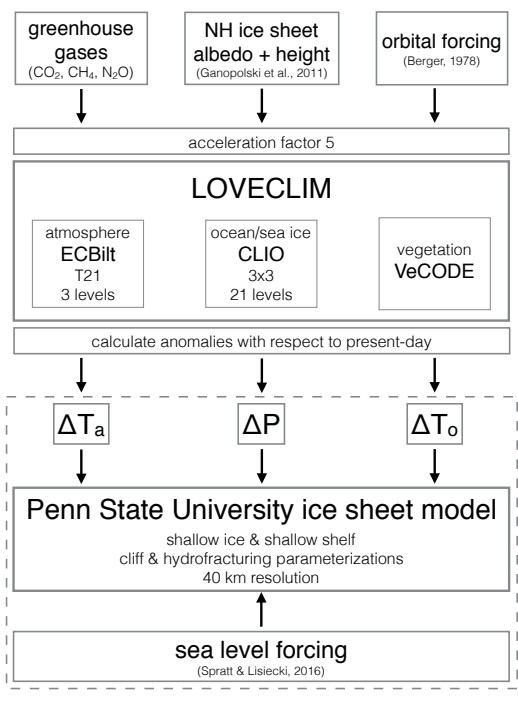

**Figure 2.** Schematic illustrating the modeling setup as described in Sect. 2.

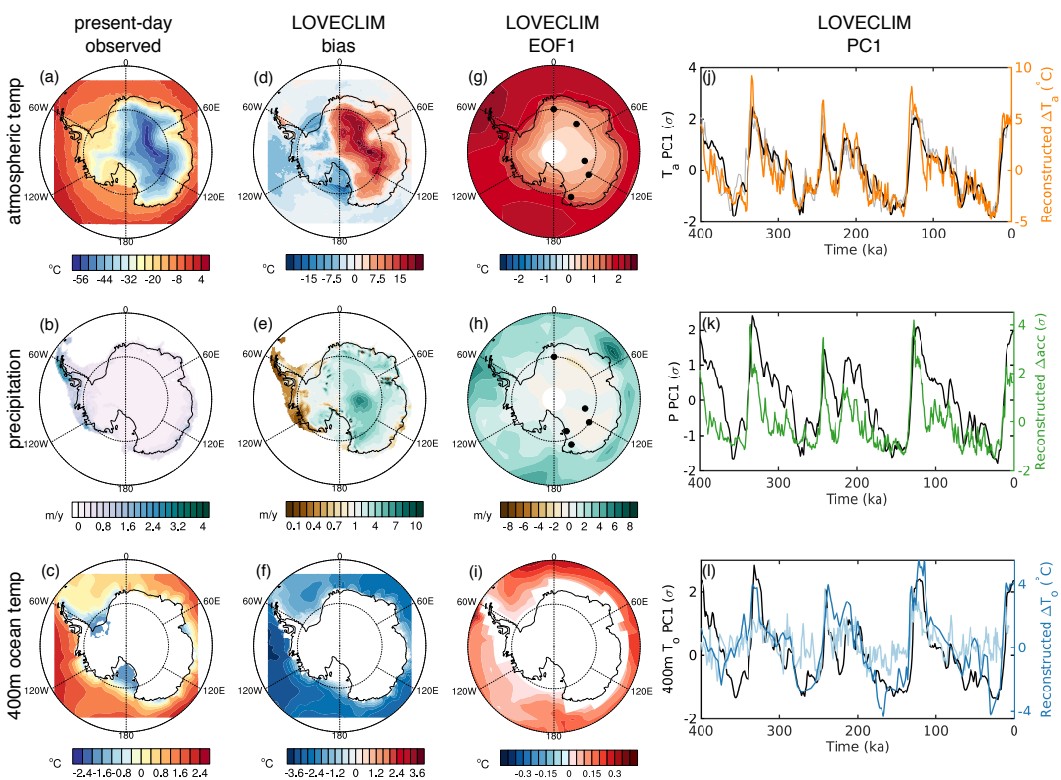

**Figure 3.** Climate forcing on the ice model grid – (left) Present-day climate conditions (Locarnini et al., 2010; Le Brocq et al., 2010), (second from left) LOVECLIM bias with respect to present-day climate, (third from left) first EOF and (right) first PC in LOVECLIM for (top) annual mean atmospheric temperature, (middle) annual mean accumulation (observed) and precipitation (LOVECLIM) and (bottom) annual mean ocean temperature at 400 m depth. Multiply the EOF pattern with the PC time series to obtain the full amplitude of the dominant mode of variability at each location. For annual mean temperature the LOVECLIM temperatures were adjusted to observed elevations (Le Brocq et al., 2010) using a lapse-rate correction of $0.008\ ^{\circ}\mathrm{C\,m^{-1}}$. The atmospheric and ocean temperature biases are plotted as LOVECLIM–observed, while the precipitation bias is plotted as LOVECLIM/observed. In addition, (j) shows a composite of reconstructed temperature anomalies from ice cores ($^{\circ}$C, orange; locations indicated by black dots in (g); Parrenin et al., 2013) and the normalized $CO_2$ record (grey; Lüthi et al., 2008), (k) shows a composite of reconstructed accumulation anomalies from ice cores ($\sigma$, green; locations indicated by black dots in (h); Steig et al., 2000; Bazin et al., 2013; Vallelonga et al., 2013) and (l) shows reconstructed deep-sea temperature anomalies at 171 $^{\circ}$W, 41 $^{\circ}$S ($^{\circ}$C, light blue; Elderfield et al., 2012) and SST anomalies at 80 $^{\circ}$W, 54 $^{\circ}$S ($^{\circ}$C, dark blue; Ho et al., 2012).

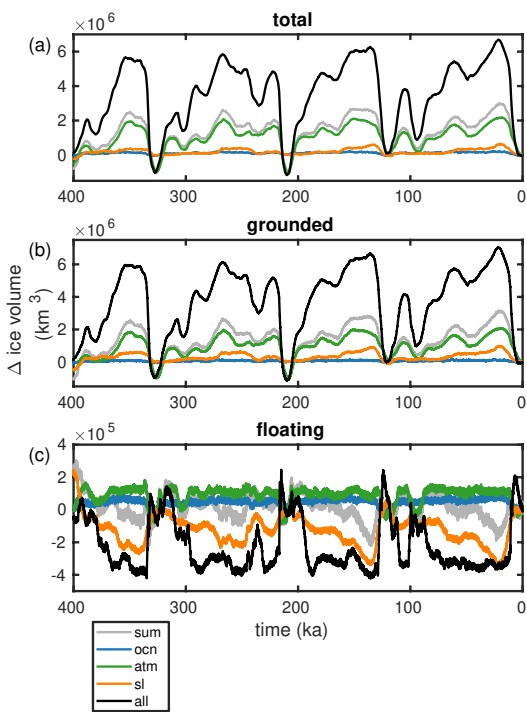

**Figure 4.** Ice sheet evolution over the last 400 ka for experiments 'ocn' (blue), 'atm' (green), 'sl' (orange), and 'all' (black) (Table 1) – (a) total ice sheet volume; (b) grounded ice sheet volume; and (c) floating ice sheet volume. The grey line is the sum of the individual runs 'ocn', 'atm', and 'sl'.

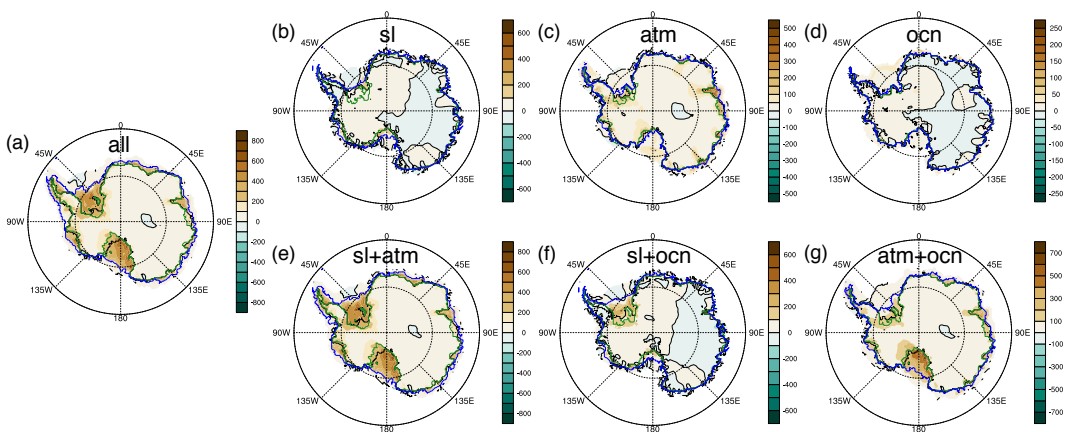

**Figure 5.** Dominant spatial pattern (first EOF) of ice sheet thickness variability (m) and minimum (green), maximum (blue) and present-day (black) grounding line extent for (a) 'all', minimum at 331 ka, maximum at 18 ka, 75.8% of variance explained; (b) 'sl', minimum at 121 ka, maximum at 18 ka, 39.4% of variance explained; (c) 'atm', minimum at 331 ka, maximum at 350 ka, 50.7% of variance explained; (d) 'ocn', minimum at 7 ka, maximum at 156 ka, 9.6% of variance explained; (e) 'sl+atm', minimum at 331 ka, maximum at 20 ka, 76.6% of variance explained; (f) 'sl+ocn', minimum at 122 ka, maximum at 140 ka, 50.5% of variance explained; (g) 'atm+ocn', minimum at 330 ka, maximum at 354 ka, 63.1% of variance explained.

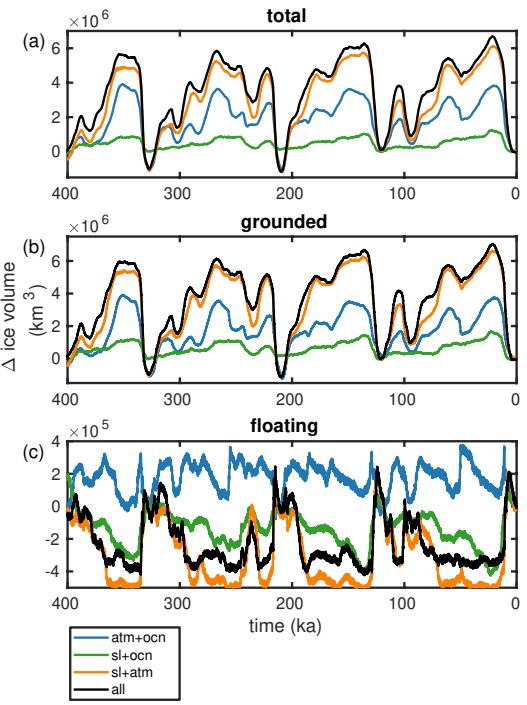

**Figure 6.** Ice sheet evolution over the last 400 ka for experiments 'atm+ocn' (blue), 'sl+ocn' (green), 'sl+atm' (orange), and 'all' (black) (Table 1) – (a) total ice sheet volume; (b) grounded ice sheet volume; and (c) floating ice sheet volume.

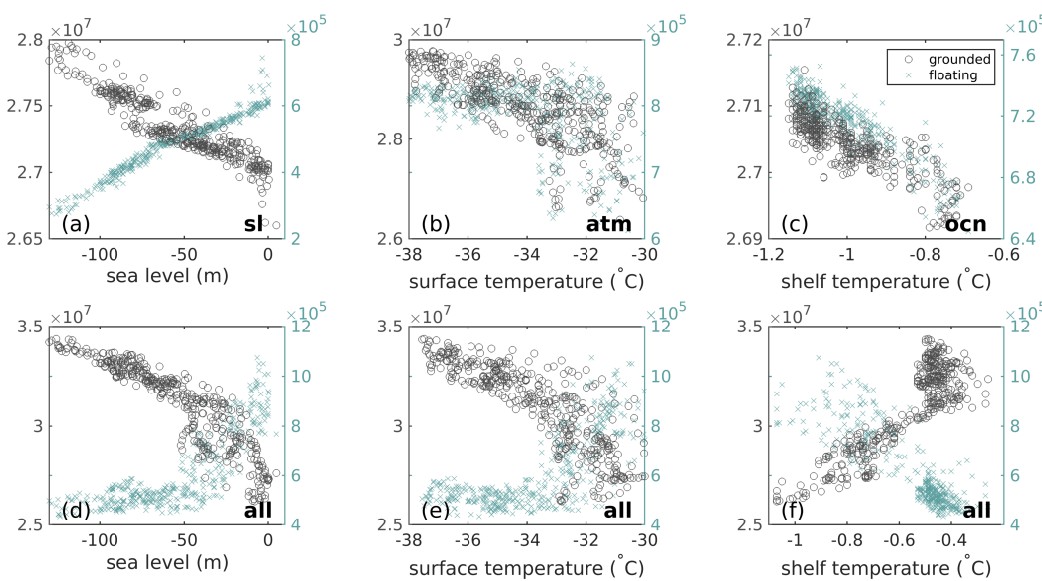

**Figure 7.** Ice sheet averaged forcing terms against floating (teal crosses) and grounded (grey circles) ice volume ($km^3$) – sea level in (a) 'sl' and (d) 'all'; atmospheric surface temperature in (b) 'atm' and (e) 'all'; and temperature beneath the ice shelves in (c) 'ocn' and (f) 'all'.

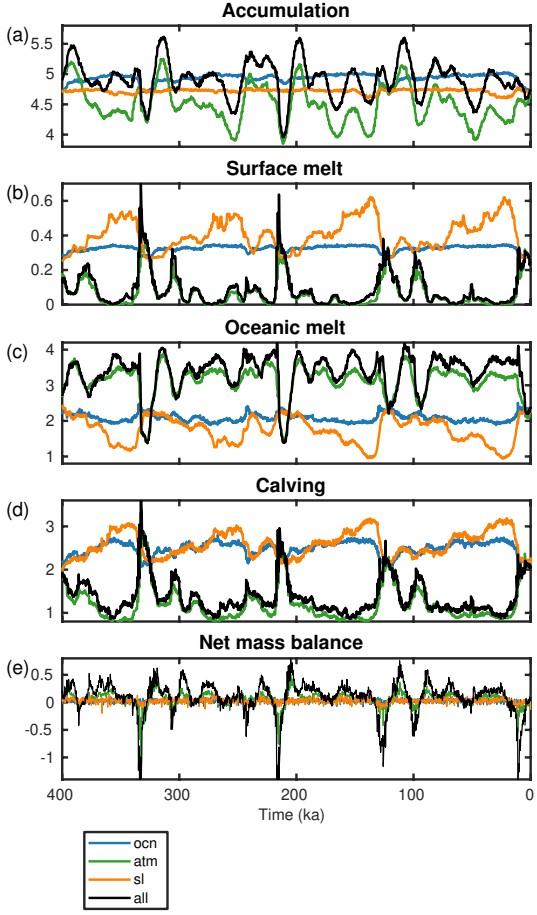

**Figure 8.** Ice sheet integrated mass balance terms ($10^3 \, \mathrm{Gt\,y^{-1}}$) for experiments 'ocn' (blue), 'atm' (green), 'sl' (orange) and 'all' (black) – (a) accumulation, (b) surface melt, (c) oceanic melt, (d) calving, and (e) net mass balance.

**Table 1.** Overview of the sensitivity experiments described in Sect. 2.2.3

| experiment | description |
| --- | --- |
| all | all forcings (Eqs. (2), (3), (4) & Spratt and Lisiecki (2016)) |
| atm | only atmospheric forcing (Eqs. (2) & (3)) |
| ocn | only ocean temperature forcing (Eq. (4)) |
| sl | only sea level forcing (Spratt and Lisiecki (2016)) |
| sl+atm | sea level and atmospheric forcing (Eqs. (2), (3) & Spratt and Lisiecki (2016)) |
| sl+ocn | sea level and ocean temperature forcing (Eqs. (4) & Spratt and Lisiecki (2016)) |
| atm+ocn | atmospheric and ocean temperature forcing (Eqs. (2), (3), (4)) |