# Peer review of "Nonlinear response of the Antarctic Ice Sheet to late-Quaternary sea level and climate forcing"

_The Cryosphere, 2019_

## Referee Comment (RC1) · Anonymous Referee #1 · 7 Jun 2019

This manuscript by Tigchelaar et al. explores the relative and combined impacts of sea level, atmosphere, and ocean forcing on Antarctic ice sheet evolution through the Quaternary. The results of their model simulations show that no individual external forcing can alone explain the entire ice sheet response, and that the forcings together exhibit a strong nonlinear response. Combined sea level and atmospheric forcing can account for most of the glacial-interglacial amplitude. Ocean melt changes are shown to be a function of ice sheet geometry rather than changes in climate. This sensitivity analysis is timely and insightful, and the manuscript is well-written. It should be of interest to readers of The Cryosphere. However, I hope the authors can address the following comments and questions. This review is divided into general and specific comments.

General comments:

1. Although this is a modelling study, the authors could reach a wider audience and better justify these experiments with more discussion of the experiments in the context of the geologic record. Much of the current debate on the relative roles of external forcings in driving Antarctic Ice Sheet changes is from surface-exposure chronologies that appear to show ice thinning of glaciers that occurs synchronously with changes in some external forcings, but not others (see Goehring et al., 2019 for a recent example). The sensitivity experiments are suited for testing these proposed mechanisms and assumptions, and this justification can be included in the introduction. It is not necessary to compare the model to every record, but a general indication of how the experiments compare to LGM reconstructions (e.g. Bentley et al., 2014) would also be of interest to many. Are the model experiments more consistent with reconstructions in some areas than others? This LGM comparison could be briefly mentioned in Section 3.2.

2. Another aspect that the authors could improve on is the clarification of caveats and model limitations, which may impact the relative and synergistic effects of the external forcings. There are two key limitations that require more detailed explanations: the sub-ice shelf melt parameterization and the sea level forcing.

2a. The relationship between ocean temperature and ice shelf depth to basal melting/freezing of ice shelves is complex and sub-ice shelf melt parameterization is an active area of research within the ice sheet model community. This is well-outlined in the review paper of Pattyn et al. (2017). The previous paper of Tigchelaar et al. (2018) offers a more detailed discussion of some of these uncertainties with respect to interglacial ocean temperatures, which is worth reiterating in this paper as well since this analysis specifically investigates the individual and combined effect of the ocean forcing. The current discussion seems too brief and there is little information offered in either paper of the parameterization used for sub-ice shelf melting/freezing (see specific comments below).

2b. Some discussion of eustatic versus relative sea level change is also warranted as relative sea level changes depend on deformational, gravitational, and rotational effects. The experiments are likely sensitive to model parameters used in the bedrock deformation relation, such as the term for the flexural rigidity of the lithosphere. It should be noted that this term has spatial variability in reality, and is quite different between East and West Antarctica. Does the ice sheet model account for this? If a single value is used, different ice sheet sectors in the model may have higher or lower relative sea level change than is realistic. This may increase or decrease the synergistic effects of the combined forcings as well. The solid Earth response has been explored in other ice sheet models with more complex bed

deformation schemes (e.g. Kingslake et al., 2018), with quite distinct ice sheet responses to external forcings with different mantle viscosity values. It is not clear if the model accounts for the gravitational or rotational components of sea level change, as in Gomez et al. (2010) and (2013). These latter components should also be discussed because gravitationally-consistent sea level change can stabilize grounding lines during periods of ice sheet retreat. This relates to the authors' conclusion that sea level forcing must be accounted for in ice sheet projections.

Specific comments:

1. Page 5, Lines 12-17: Please show the equation for the sub-ice shelf melt parameterization. Based on the reference provided, I assume that it is Eq. 17 in Pollard and DeConto (2012). If not, please clarify. If so, what is the value used for the transfer factor ($K_T$) and what is it based on? Are different K values used in different basins? How sensitive are ice shelf melt/freeze rates to the value of this parameter in relation to the ocean temperature anomalies? Are modelled melt/freeze rates reasonable with present-day climate forcing?

2. Page 5, Lines 25-29: It should be noted that the MICI parameterization is still a topic of debate, and it may not be necessary to reproduce Antarctic sea level contributions of past warm periods (see Edwards et al., 2019).

3. Page 6, Line 24-25: With the caveat that 34.5 psu may not be appropriate for the ocean salinity at the ice-ocean interface.

4. Page 7, Line 25: Can the authors clarify the purpose of including the EOF1 plots in Figure 3?

5. Page 8, Line 12: The benthic Southern Ocean temperature reconstruction of Elderfield et al. (2012) could also be included as an alternative to the SST reconstruction.

6. Page 9, Line 31: Remove the comma before "because"

7. Page 10, Line 1: Remove the comma before "because"

8. Page 12, Lines 15-26: Meltwater fluxes may partly explain the low ocean temperature anomalies in the climate model, but I would also add that some of this may be specific to LOVECLIM. The authors previously mention the model overestimates present-day minimum sea ice extent and that this may contribute to the underestimation of ocean temperatures (Page 7, Lines 3-4). Glacial ocean temperature anomalies are much more negative in CCSM3 along the Antarctic coasts than in LOVECLIM (see Lowry et al., 2018). Other climate models may also show more positive ocean temperature anomalies during interglacial periods than LOVECLIM.

9. Figure 1: A darker colour for $CO_2$ and obliquity would make the axes easier to read.

10. Figure 3: Why do the temperature/accumulation composites include only East Antarctic ice cores? Please clarify the spatial domain for the PC1 lines in panels j-l.

11. Figure 7: See above comment for Figure 1.

12. Figure 8: Move the legend outside the plot so that it is not overlying the mass balance curves.

References

Bentley, M. J., Cofaigh, C. O., Anderson, J. B., Conway, H., Davies, B., Graham, A. G., ... & Mackintosh, A. (2014). A community-based geological reconstruction of Antarctic Ice Sheet deglaciation since the Last Glacial Maximum. *Quaternary Science Reviews*, *100*, 1-9.

Edwards, T. L., Brandon, M. A., Durand, G., Edwards, N. R., Golledge, N. R., Holden, P. B., ... & Wernecke, A. (2019). Revisiting Antarctic ice loss due to marine ice-cliff instability. *Nature*, *566*(7742), 58.

Elderfield, H., Ferretti, P., Greaves, M., Crowhurst, S., McCave, I. N., Hodell, D. A., & Piotrowski, A. M. (2012). Evolution of ocean temperature and ice volume through the mid-Pleistocene climate transition. *Science*, *337*(6095), 704-709.

Goehring, B. M., Balco, G., Todd, C., Moening-Swanson, I., & Nichols, K. (2019). Late-glacial grounding line retreat in the northern Ross Sea, Antarctica. *Geology*, *47*(4), 291-294.

Gomez, N., Mitrovica, J. X., Huybers, P., & Clark, P. U. (2010). Sea level as a stabilizing factor for marine-ice-sheet grounding lines. *Nature Geoscience*, *3*(12), 850.

Gomez, N., Pollard, D., & Mitrovica, J. X. (2013). A 3-D coupled ice sheet–sea level model applied to Antarctica through the last 40 ky. *Earth and Planetary Science Letters*, *384*, 88-99.

Kingslake, J., Scherer, R. P., Albrecht, T., Coenen, J., Powell, R. D., Reese, R., ... & Whitehouse, P. L. (2018). Extensive retreat and re-advance of the West Antarctic Ice Sheet during the Holocene. *Nature*, *558*(7710), 430.

Lowry, D. P., Golledge, N. R., Menviel, L., & Bertler, N. A. (2019). Deglacial evolution of regional Antarctic climate and Southern Ocean conditions in transient climate simulations. *Climate of the Past*, *15*(1), 189-215.

Pattyn, F., Favier, L., Sun, S., & Durand, G. (2017). Progress in numerical modeling of Antarctic ice-sheet dynamics. *Current Climate Change Reports*, *3*(3), 174-184.

Pollard, D., & DeConto, R. M. (2012). Description of a hybrid ice sheet-shelf model, and application to Antarctica. *Geoscientific Model Development*, *5*(5), 1273-1295.

Tigchelaar, M., Timmermann, A., Pollard, D., Friedrich, T., & Heinemann, M. (2018). Local insolation changes enhance Antarctic interglacials: Insights from an 800,000-year ice sheet simulation with transient climate forcing. *Earth and Planetary Science Letters*, *495*, 69-78.

---

## Referee Comment (RC2) · Johannes Sutter (Referee) · 28 Jun 2019

Review of Tigchelaar et al. :

„Nonlinear response of the Antarctic ice sheet to Quaternary sea level and climate forcing"

Tigchelaar et al. discuss the transient evolution of the Antarctic Ice Sheet in response to late Quaternary climatological and sea level boundary conditions by means of 3D ice sheet modelling. They force their ice sheet model with the output of an Earth System Model of intermediate complexity covering the last 408 kyr employing a continuous sea level forcing. Their discussion of the individual impact of a single forcing component versus the integrated effect of combined forcings on ice volume and geometry changes is intriguing and worth publishing. The topics covered in their manuscript fit into the scope of The Cryosphere and the manuscript is generally well written. In principal I would suggest publication of their manuscript if some key issues regarding the discussion of their results and methodology are addressed.

In the following I will lay out my main general remarks which should be addressed before publication followed by some minor corrections of typos and wording.

1. Methods and ocean forcing.

While the method section is clearly written, I think that a more transparent discussion of the strengths and weaknesses of the forcing approach would improve the manuscript and help the reader to put the results into perspective with the literature in the field. The fact that a transient model run spanning several hundred thousand years is used to force the ice sheet model is very impressive, but it should be clearly stated that this is at the expense of resolution which is very coarse. It is for example well known that global ocean models have a hard time resolving circumantarctic circulation which mostly leads to inaccurate representations of warming during interglacials and therefore a muted response of the Antarctic Ice Sheet (e.g. Sutter et al., 2016). I would imagine that the representation of variability of circumantarctic ocean temperatures is even worse in EMICs. I guess this is one of the reasons why ice sheet volume remains relatively high for most interglacials in the manuscript presented here, as well as in Tigchelaar et al. (2018). Throughout the manuscript (Methods, Results, Discussion), it should be re-iterated that with the ocean forcing used in this manuscript, the impact of ocean temperatures on transient interglacial ice sheet dynamics in the late Quaternary cannot be accurately assessed. Upon reading the manuscript, I had the impression that the results shown here imply that ocean temperature forcing in Interglacials or deglaciation phases is not important for ice sheet retreat which would contradict the current literature on how the Antarctic Ice Sheet responds to warmer worlds or in glacial terminations. I am sure that this is not the intended take away message but the chance of misinterpretation for someone not familiar with the field is high.

2. Ice sheet model

The description of the ice sheet model is rather compact, which is probably due to the fact, that it has been discussed at length in the cited literature. However, a quick reference as how ice shelf mass balance is treated would be helpful (calving, basal melt parameterization). Providing an assessment of how well the basal ice shelf melt

pattern matches the present day observed melt rate (e.g. Depoorter et al., 2013, Rignot et al., 2013) would be helpful as well. How is the model tuned, and how does it perform against present day and paleo benchmarks? Also it would be worth mentioning how the resolution (40 km) used here could affect the results compared to higher resolution studies.

3. Representation of ocean temperatures

The authors mention in section 2.2.2, that LOVECLIM Southern Ocean temperatures are generally too coold. As you use an anomaly forcing to prevent bias propagation it would be interesting how big the glacial and interglacial temperature anomalies (e.g. at 400 m depth) close the the ice shelves are.

4. Presentation of Results

The presentation and discussion of the results is currently written in a very affirmative manner which sometimes ignores the biases introduced by the experimental setup. For example 3.4.4 suggests that temporal ocean temperature changes are not relevant for ice volume changes. While this is true for the setup used here, it is not the case in multiple publications on the matter (e.g. Golledge et al., 2015, 2017, 2019, DeConto & Pollard 2016, Sutter et al. 2016,2019, Albrecht et al., 2019 TCD). The authors state that they will discuss the validity of the results in the Discussion (p.11 L 10-11), but I have the impression that a serious debate about the shortcomings of the approach and therefore the scope of the results is lacking.

Specific comments:

Title: as the authors focus on the AIS evolution during the last 408 ka I would rename the title to

"Nonlinear response of the Antarctic ice sheet to **late-**Quaternary sea level and climate forcing"

and use late-Quaternary instead of Quaternary throughout the manuscript (already done in the header of section 3.1).

Check Antarctic Ice Sheet (AIS) throughout the manuscript. Usually it is written in capital initial letters. Furthermore, while you introduce the abbreviation on page 1 L 17 you mostly don't use it later on.

P2 L10: The importance for what?

P2 L18-19: I think this is mostly true for glacials but less so for interglacials. While e.g. Konrad et al. (2014) show that a sea level drop due to changes in the gravitational pull during ice loss in interglacials can stabilize the grounding line the overall rise in sea level during interglacials doesn't play a large role in grounding line retreat as it is mostly limited to just a few meters.

P2 L22: maybe rephrase to : "particularly leading to a growth of the East Antarctic Ice Sheet (EAIS)"

P3 L11-14: inconsistent use of Section versus Sect.

P4 L1: maybe rephrase to "Each land grid cell …"

P4 L19: rephrase to "While the cimate model run closely follows …, here the longwave radiative effect of CO2 was amplified …"

P4 L34: is the duration of the experiments the reason for the 40km resolution? Then rephrase: Due to limited computational resources and long timescale of the simulations we had to use a relatively coarse resolution of 40 km.

P5 L1: rephrase to "Present day climate forcing is obtained from the […] interpolated to the ice model grid."

Do you use the ALBMAP v1 bedrock topography or BEDMAP2 for the initial ice sheet configuration?

P5 L12-30: I would expect the description of the parameterization of the basal shelf melt calculation and calving to be in the ice sheet model section and not in the climate forcing section.

P6 L6-8: Here I do not understand whether the climate forcing "jumps" every thousand years to a new set of climate anomalies (i.e. the ISM is forced with the same climate anomalies for 1000 years) or whether the transition is smooth. Please clarify.

P6 L 8-10: maybe rephrase to:

The atmospheric temperature $T_a$ is modified by a lapse rate correction of $\gamma=0.008°Cm^{-1}$ to account for surface elevation differences between the reference ice sheet geometry ($z^{obs}$; Le Brocq et al., 2010) and both the simulated elevation at time t ($z(t)$), as well as for differences with respect to the LOVECLIM orography ($z^{LC}$).

P7 L1-2: As you force the ISM with ocean temperature anomalies I guess the glacial-interglacial variability is more relevant for the ice sheet's evolution than the present day bias. Please add a sentence which quantifies the ocean warming e.g. in MIS5e and MIS11 and the cooling e.g. during the LGM relative to the PI control climate state (LOVECLIM 1000 year average?).

P7 L 11-12: remove sentence "The bottom half of Fig. 2 …"

P7 L26: it would help the reader if $CO_2$ is plotted in Fig. 3j as well to make the pacing more evident.

P7 L31: Maybe I overlooked this but how do you create the ice core composite? If I understand it correctly you use Dome Fuji, EDC, Vostok, TALDICE and EDML. Only Dome Fuji, EDC, Vostok cover the whole 408 ka.

P8 L8: Again, you use only one coastal ice core (TALDICE) and 4 interior ice cores. For the latter, the lapse rate correction would be stronger in glacials (e.g. Pollard et

al. 2009 and Sutter et al. 2019 TCD Fig. 10). But the biggest discrepancies shown in 3l and 3j occur in Interglacials with too cold ocean and surface temperatures.

P8 L17-18: I could imagine that the underestimation of ocean temperature variability in interglacials is the main reason why the ocean forcing is the weakest driver of interglacial ice volume changes in your simulations. This has important implications for your conclusions as this is a methodological bias and not necessarily reflects the actual response of the AIS e.g. in MIS5e and MIS11.

P8, L23-26: Actually MIS7 shows the lowest surface temperature warming in Antarctic ice cores, how come that for this period the AIS volume is higher than in the other interglacials in your simulations?

P9, L12: rephrase to: Figure 5 shows where the individual forcing components have the largest effect on the Antarctic Ice Sheet.

P9, L14-15: what is the reason for this thickening? Reduced surface melt? Retreat caused by hydrofracturing?

P9, L16: again I expect this to be caused by the forcing setup and that it is not representative during Interglacials.

P9, L20: maybe rephrase to: Combined forcing leads to a more pronounced grounding line advance during glacials than in simulations with single forcing.

P9, L27: rephrase to: Figure 7 depicts the response of grounded ice volume to the respective forcing in the different sensitivity runs.

P9, L27: rephrase to: It is important to note that the impact of the sea level forcing in isolation leads to the conversion of grounded into floating ice (during Terminations???).

P9, L30: Do surface melt rates really increase in glacials?? The maximum elevation change of ice shelves during glacials would be ca. 120 m.

P10, L5: quantify "fairly consistently".

P10, L6: Wording. (Now, …)

P10, section 3.4.3. This section needs to be expanded, discussing the reasons why the ocean forcing plays a negligible role in the simulations (see main remarks).

P10, L23: rephrase to: Our sensitivity runs show that the **simulated** response of the AIS to **late** Quaternary external drivers […]

Section 3.4.4. and the Discussion requires a more detailed disentanglement of what the authors deem to be realistic responses of the AIS to late Quaternary climate and boundary conditions and what they think is due to methodological biases.

P11, L30: I do not understand this sentence. Increased ice loss due to sea level rise induced warming? This needs to more explicit, warming due to sea level driven ice

sheet retreat and therefore surface lowering?

P12, L7-8: What is meant by "manually offset"?

P12, L10: rephrase, e.g. : In particular, as the ice sheet grows ice sheet areas with higher precipitation expand leading to a positive feedback while at the same time, the ice margin advances into warmer ocean waters which leads to a negative feedback.

P12, L15-18: This is the only place in the manuscript which states that LOVECLIM ocean temperature variability is too low and that this could be causal to the muted response during interglacials. Unfortunately, this sentence is right away relativized in the next sentence, citing **one** publication, while a wealth of publications identified ocean warming to be the main driver of ice loss in late Quaternary interglacials (e.g. Golledge et al., 2015,2017,2019, DeConto & Pollard 2016, Sutter et al., 2016).

P12, L28-29: replace sentence "Previous modeling studies have failed to elucidate how these different external drivers interact in driving large glacial ice sheet growth and interglacial sea level highstands." E.g. With "In contrast to previous studies, here we focus on the interaction of different external forcings driving Antarctic Ice Sheet changes". There are previous studies who discuss individual forcing components (e.g. Pollard et al. 2009, deBoer et al. 2013), just not as comprehensive as done here.

P12, L 33-35: I am not fully convinced that this is the case, or at least that this study shows that, as the effect of ocean temperature changes in interglacial ice sheet retreat are not adequately captured in the simulations presented here.

Figures 1,4,5,6,8 : move labels a),b),c) out of the figures panels

References:

(Pollard 2009, de Boer, van de Wal et al. 2013, Depoorter, Bamber et al. 2013, Rignot, Jacobs et al. 2013, Konrad, Thoma et al. 2014, Golledge, Kowalewski et al. 2015, DeConto and Pollard 2016, Sutter, Gierz et al. 2016, Golledge, Levy et al. 2017, Tigchelaar, Timmermann et al. 2018, Golledge, Keller et al. 2019)

de Boer, B., R. S. W. van de Wal, L. J. Lourens, R. Bintanja and T. J. Reerink (2013). "A continuous simulation of global ice volume over the past 1 million years with 3-D ice-sheet models." Climate Dynamics **41**(5-6): 1365-1384.
DeConto, R. M. and D. Pollard (2016). "Contribution of Antarctica to past and future sea-level rise." Nature **531**(7596): 591-597.
Depoorter, M. A., J. L. Bamber, J. A. Griggs, J. T. M. Lenaerts, S. R. M. Ligtenberg, M. R. van den Broeke and G. Moholdt (2013). "Calving fluxes and basal melt rates of Antarctic ice shelves." Nature **502**(7469): 89-93.
Golledge, N. R., E. D. Keller, N. Gomez, K. A. Naughten, J. Bernales, L. D. Trusel and T. L. Edwards (2019). "Global environmental consequences of twenty-first-century ice-sheet melt." Nature **566**(7742): 65-+.

Golledge, N. R., D. E. Kowalewski, T. R. Naish, R. H. Levy, C. J. Fogwill and E. G. W. Gasson (2015). "The multi-millennial Antarctic commitment to future sea-level rise." Nature **526**(7573): 421-+.

Golledge, N. R., R. H. Levy, R. M. McKay and T. R. Naish (2017). "East Antarctic ice sheet most vulnerable to Weddell Sea warming." Geophysical Research Letters **44**(5): 2343-2351.

Konrad, H., M. Thoma, I. Sasgen, V. Klemann, K. Grosfeld, D. Barbi and Z. Martinec (2014). "The Deformational Response of a Viscoelastic Solid Earth Model Coupled to a Thermomechanical Ice Sheet Model." Surveys in Geophysics **35**(6): 1441-1458.

Pollard, D. D., R. M. (2009). "Modelling West Antarctic ice sheet growth and collapse through the past five million years." Nature **458**: 329-333.

Rignot, E., S. Jacobs, J. Mouginot and B. Scheuchl (2013). "Ice-Shelf Melting Around Antarctica." Science **341**(6143): 266-270.

Sutter, J., P. Gierz, K. Grosfeld, M. Thoma and G. Lohmann (2016). "Ocean temperature thresholds for Last Interglacial West Antarctic Ice Sheet collapse." Geophysical Research Letters **43**(6): 2675-2682.

Tigchelaar, M., A. Timmermann, D. Pollard, T. Friedrich and M. Heinemann (2018). "Local insolation changes enhance Antarctic interglacials: Insights from an 800,000-year ice sheet simulation with transient climate forcing." Earth and Planetary Science Letters **495**: 69-78.

---

## Author Comment (AC1) · 18 Aug 2019

The authors thank the anonymous reviewer for their thorough review and helpful comments. A full response to their review and that of reviewer #2 is attached as a supplement.

Please also note the supplement to this comment:
https://www.the-cryosphere-discuss.net/tc-2019-83/tc-2019-83-AC1-supplement.pdf

---

## Author Comment (AC2) · 18 Aug 2019

*"Nonlinear response of the Antarctic Ice Sheet to
late-Quaternary sea level and climate forcing"*

**Author response to reviewer comments**

The authors thank Johannes Sutter and an anonymous reviewer for their thorough review and helpful comments. Below is a line-by-line response to their feedback, with reviewer comments in italics.

**Reply to Reviewer Comment 1 (Anonymous)**

*General comments:*

*1. Although this is a modelling study, the authors could reach a wider audience and better justify these experiments with more discussion of the experiments in the context of the geologic record. Much of the current debate on the relative roles of external forcings in driving Antarctic Ice Sheet changes is from surface-exposure chronologies that appear to show ice thinning of glaciers that occurs synchronously with changes in some external forcings, but not others (see Goehring et al., 2019 for a recent example). The sensitivity experiments are suited for testing these proposed mechanisms and assumptions, and this justification can be included in the introduction. It is not necessary to compare the model to every record, but a general indication of how the experiments compare to LGM reconstructions (e.g. Bentley et al., 2014) would also be of interest to many. Are the model experiments more consistent with reconstructions in some areas than others? This LGM comparison could be briefly mentioned in Section 3.2.*

Thank you for this excellent suggestion. With regards to contextualizing the motivation for these simulations with the deglaciation record, we will add a few sentences about this to the introduction. As for a comparison against the LGM reconstructions, this was discussed in more detail in Tigchelaar et al. (2018). There we wrote that "During glacial maxima, the AIS grounding line extends to the continental shelf break almost everywhere (Fig. 2b). The simulated local LGM ice volume maximum occurs from 23–20 ka, with the grounding line position at this time generally in close agreement with reconstructions. The last deglaciation begins in the Bellingshausen sector before 16 ka, followed by the Amundsen sector around 13 ka, and the Weddell, Ross and Amery sectors around 10 ka and beyond (SI Fig. 2), a pattern in general agreement with time slice reconstructions of grounding line position (Bentley et al., 2014). However, our retreat in the Ross sector occurs at least ~2kyr earlier, and further work will focus on this discrepancy. In the Ross sector, there is an "overshoot" of Siple Coast grounding lines at ~6 to 4ka, and a subsequent re-advance to the modern positions by 0ka. Similar but more localized behaviour occurs in the Weddell sector (SI Fig. 2). This feature was also described in Maris et al. (2014), and is probably due to time-lagged isostatic bedrock rebound and shallowing grounding lines allowing re-advance in the late Holocene (Bradley et al., 2015)."

As mentioned there, the challenge to achieve good simulations of the last deglacial retreat remains a major focus of our and others' modeling efforts. We and others have

applied large ensembles of model parameter sets, and automated scoring algorithms (Briggs et al., 2013, 2014; Pollard et at 2016, 2017), comparing with several diverse types of paleo data, reviewed for instance in RAISED (2014). This data includes grounding-line positions vs. time, and cosmogenically-derived thinning in inland marginal areas. We will add a brief discussion with some of these comparisons to Sect. 3.2 of the manuscript.

*2. Another aspect that the authors could improve on is the clarification of caveats and model limitations, which may impact the relative and synergistic effects of the external forcings. There are two key limitations that require more detailed explanations: the sub-ice shelf melt parameterization and the sea level forcing.*

*2a. The relationship between ocean temperature and ice shelf depth to basal melting/freezing of ice shelves is complex and sub-ice shelf melt parameterization is an active area of research within the ice sheet model community. This is well-outlined in the review paper of Pattyn et al. (2017). The previous paper of Tigchelaar et al. (2018) offers a more detailed discussion of some of these uncertainties with respect to interglacial ocean temperatures, which is worth reiterating in this paper as well since this analysis specifically investigates the individual and combined effect of the ocean forcing. The current discussion seems too brief and there is little information offered in either paper of the parameterization used for sub-ice shelf melting/freezing (see specific comments below).*

This is a very valid criticism and one that was shared by the other reviewer. As detailed below, we will expand the discussion of the basal melt parameterization in the Methods section. We will also be more explicit about the shortcomings of the ocean temperature forcing used, and the implications for conclusions about relative importance of external drivers. Specifically, we will elaborate on this in Sect. 3.1 (climate forcing), 3.4.3 (ocean-only run), 3.4.4 (combined forcings), and the Discussion.

For the Discussion, we suggest the following updated text:
"Our finding that ocean temperature forcing plays a limited role in driving changes in Antarctic ice volume contrasts with previous modeling studies of past and future AIS evolution (Golledge et al., 2015; DeConto and Pollard, 2016; Sutter et al., 2016), as well as observations of sustained sub-shelf ice loss in response to ongoing ocean warming at e.g., Pine Island Glacier (Jacobs et al., 2011; Pritchard et al., 2012). This is not surprising, given that the LOVECLIM-simulated ocean temperature anomalies are small (Figs. 3i,l), and ice sheet models typically need ocean warming of 2-5 °C to initiate interglacial WAIS collapse (Pollard and DeConto, 2009; DeConto and Pollard, 2016; Sutter et al., 2016; Tigchelaar et al., 2018). Absent paleo-reconstructions of near-Antarctic sub-surface ocean temperatures it is difficult to assess how realistic our LOVECLIM simulation is, though critical processes such as Antarctic Bottom Water formation are known to be poorly represented in low-resolution climate models (e.g., Snow et al., 2015), and previous studies have found LOVECLIM in particular to have more muted late-Quaternary temperature variability than other models (Lowry et al., 2019).

In addition, many regional oceanographic processes can affect the circum-Antarctic ocean environment beyond large-scale climate forcing. For example, the blocking effects of sea ice formation (Hellmer et al., 2012), the role of winds in pushing warm waters onto the continental shelf (Thoma et al., 2008; Steig et al., 2012), and the complex geometry of ice shelf cavities (Jacobs et al., 2011; De Rydt et al., 2014) have all been found to be important in observational and modeling studies of current and future oceanic melting of the WAIS ice shelves (Joughin et al., 2014). For that reason, using 400m-depth Southern Ocean temperatures as the sole driver for sub-shelf melt may miss important near-Antarctic dynamics. Furthermore, melt water fluxes from the AIS have been found to lead to cooling of surface waters and warming at intermediate depth (Menviel et al., 2010; Weber et al., 2014), a feedback mechanism that could increase ice sheet loss (Golledge et al., 2014, 2019). These processes can only really be captured in fully coupled ocean-atmosphere-ice sheet simulations at high resolution, something that is currently not feasible for the long timescales of late-Quaternary climate evolution. However, it should be possible to run shorter simulations – of e.g., the Last Interglacial or Marine Isotope Stage 11 – using such a setup, and perform a similar set of sensitivity experiments as done here. This would likely reveal additional nonlinearities as ice sheet and forcing are allowed to evolve together. The accumulation forcing for instance is similarly impacted by low climate model resolution and lack of ice-climate feedbacks. Time-evolving changes in orography and albedo can substantially alter atmospheric circulation patterns and associated rainfall (Steig et al., 2000; Maris et al., 2014; Steig et al., 2015)."

*2b. Some discussion of eustatic versus relative sea level change is also warranted as relative sea level changes depend on deformational, gravitational, and rotational effects. The experiments are likely sensitive to model parameters used in the bedrock deformation relation, such as the term for the flexural rigidity of the lithosphere. It should be noted that this term has spatial variability in reality, and is quite different between East and West Antarctica. Does the ice sheet model account for this? If a single value is used, different ice sheet sectors in the model may have higher or lower relative sea level change than is realistic. This may increase or decrease the synergistic effects of the combined forcings as well. The solid Earth response has been explored in other ice sheet models with more complex bed deformation schemes (e.g. Kingslake et al., 2018), with quite distinct ice sheet responses to external forcings with different mantle viscosity values. It is not clear if the model accounts for the gravitational or rotational components of sea level change, as in Gomez et al. (2010) and (2013). These latter components should also be discussed because gravitationally-consistent sea level change can stabilize grounding lines during periods of ice sheet retreat. This relates to the authors' conclusion that sea level forcing must be accounted for in ice sheet projections.*

In this work, as in many previous comparable studies, we use a standard ELRA (Elastic Lithosphere Relaxing Asthenosphere) model for bed depression and rebound under the varying ice load. Other much more complex and comprehensive full Earth models are available, and have previously been used in shorter time-scale runs (e.g., Gomez et al., 2013, 2015, 2018; Pollard et al., 2017) but coupling with our full-Earth model setup

would be computationally prohibitive for the 400-kyr time scales of the simulations here. These previous studies have addressed the sensitivity of Antarctic results to ELRA vs. full-Earth models in shorter experiments (last deglacial since 20 ka, and future ~3 kyr). There are two feedbacks introduced by the full-Earth models, due to (i) ice-ocean gravitational interaction, and (ii) low-viscosity mantle zone over regions of West Antarctica. Both of these are potentially negative feedbacks for ice retreat, as they cause shallower ocean depths at rapidly retreating grounding lines and thus smaller ice flux from the interior.

However, these studies have found that the last-deglacial and future retreats in Antarctic basins are not strongly affected, at least for standard Earth structures using the full-Earth model compared to ELRA; also a weak low-viscosity mantle zone only has significant effects for very strong forcing such as future business-as-usual emission scenarios, and not for slower past glacial-interglacial forcing (Gomez et al., 2013, 2015; Pollard et al., 2017). Consequently, we suggest that full-Earth coupling would have only small effects on the paleoclimatic results here. However, it should be addressed in future work, when coupling with full-Earth models in our system becomes computationally feasible for near 1-Myr run lengths. Developments to allow 3D variations in Earth structure, and also to allow much longer run lengths, are in progress (Gomez et al., 2018).

We will address this in the paper by adding the following sentences to the last paragraph of Methods Section 2.2.2: "The PSU-ISM uses a standard Elastic Lithosphere Relaxing Asthenosphere model for bed depression and rebound under the varying ice load, and therefore does not include deformational, gravitational, and rotational contributions to local sea level change. Such contributions would potentially act as negative feedbacks for ice retreat, and cause spatial variability between the East and West Antarctic ice sheets (Gomez et al., 2015). However, previous work that includes full-Earth coupling suggest this would likely only have small effects for our timescales (Gomez et al., 2013, 2015; Pollard et al., 2017), and it is currently not computationally feasible to run a full-Earth model for our 400 kyr simulations (though work is in progress to improve this, e.g., Gomez et al., 2018)."

We will also add a sentence to the second paragraph of the discussion: "Such future studies should make sure to include the deformational and gravitational components of future sea level rise through coupling with a full-Earth model (Gomez et al., 2018)."

*Specific comments:*

*1. Page 5, Lines 12-17: Please show the equation for the sub-ice shelf melt parameterization. Based on the reference provided, I assume that it is Eq. 17 in Pollard and DeConto (2012). If not, please clarify. If so, what is the value used for the transfer factor (KT) and what is it based on? Are different K values used in different basins? How sensitive are ice shelf melt/freeze rates to the value of this parameter in relation to the ocean temperature anomalies? Are modelled melt/freeze rates reasonable with present-day climate forcing?*

The parameterization of oceanic basal melting under ice shelves uses Eq. 17 in Pollard and DeConto (2012), but has been consolidated with no ad-hoc variations in coefficients (i.e., K=3 everywhere), as described in Pollard et al. (2015, Supplemental Information). As in several other models, the melt rate is proportional (with a single coefficient) to the square of the temperature difference between the base of the ice and the closest grid point at 400 m depth in an ocean dataset. This yields reasonable patterns of modern sub-ice shelf melt, as shown in the Appendix of Pollard et al. (2017), which are generally within ongoing estimates of empirical uncertainties and rapidly changing decadal trends (Depoorter et al., 2013., Rignot et al, 2013). As a design principle, a spatially uniform coefficient is used rather than tuning it point-by-point to match modern estimated maps, because ocean circulation and the associated best-fit coefficients could well change drastically as cavity geometries underneath ice shelves vary over the course of past or future long-term simulations. We will include the equation, parameter values, and a brief discussion of comparison against observations in the manuscript.

*2. Page 5, Lines 25-29: It should be noted that the MICI parameterization is still a topic of debate, and it may not be necessary to reproduce Antarctic sea level contributions of past warm periods (see Edwards et al., 2019).*

This is true, and we will make a note of this in the paper. More pertinently, the MICI parameterization is not triggered in our simulations, because interglacial climate anomalies simulated by LOVECLIM are only marginally warmer than present-day temperatures (Fig. 3j). We will add the following to the manuscript: "It is worth noting that these parameterizations – whose validity continues to be debated (Edwards et al., 2019) – do not get triggered by our simulated late-Quaternary climate anomalies (Sect. 3.1)."

*3. Page 6, Line 24-25: With the caveat that 34.5 psu may not be appropriate for the ocean salinity at the ice-ocean interface.*

The value used (34.5 psu) is selected to represent depths of ~400 m – typical depths of Circumpolar Deep Water (CDW) – but would admittedly be lower in the presence of ice melt. The value for the ocean salinity used in the basal melting parameterization only enters in setting the freezing point and has a very small effect on the basal melt rates, especially compared to uncertainties in water temperatures in the ocean dataset. We will make a note of this in the manuscript.

*4. Page 7, Line 25: Can the authors clarify the purpose of including the EOF1 plots in Figure 3?*

We include the EOF1 plots because in order to retrieve the local amount of change captured by this mode, one needs to multiply the spatial pattern (EOF) with the time series (PC). We will add a sentence in the first paragraph of Sect. 3.1 to clarify to the reader how these figures can be used together to infer spatio-temporal variability in the climate data: "The full amplitude of this first mode at each point in space can be derived

by multiplying the EOF1 map with the PC1 time series." We will add a similar note to the caption of Fig. 3.

*5. Page 8, Line 12: The benthic Southern Ocean temperature reconstruction of Elderfield et al. (2012) could also be included as an alternative to the SST reconstruction.*

Thank you for this suggestion. Neither the benthic record nor the SST reconstructions will fully capture the processes driving temperature changes at intermediate water depth, but including them both will give a sense of the magnitude of the differences from surface to bottom. We will include the Elderfield reconstruction in Fig. 3l.

*6. Page 9, Line 31: Remove the comma before "because"*

This will be done.

*7. Page 10, Line 1: Remove the comma before "because"*

This will be done.

*8. Page 12, Lines 15-26: Meltwater fluxes may partly explain the low ocean temperature anomalies in the climate model, but I would also add that some of this may be specific to LOVECLIM. The authors previously mention the model overestimates present-day minimum sea ice extent and that this may contribute to the underestimation of ocean temperatures (Page 7, Lines 3-4). Glacial ocean temperature anomalies are much more negative in CCSM3 along the Antarctic coasts than in LOVECLIM (see Lowry et al., 2018). Other climate models may also show more positive ocean temperature anomalies during interglacial periods than LOVECLIM.*

Thank you. We will update this part of the discussion to emphasize that both LOVECLIM itself and the chosen model setup (low-resolution EMIC not coupled to ice-sheet model) contribute to errors in estimating the sub-shelf ocean temperature anomalies.

*9. Figure 1: A darker colour for CO2 and obliquity would make the axes easier to read.*

This will be done.

*10. Figure 3: Why do the temperature/accumulation composites include only East Antarctic ice cores? Please clarify the spatial domain for the PC1 lines in panels j-l.*

The temperature and accumulation time series were chosen for their length, to better allow for comparison against the long simulation with the climate model. All chosen EAIS cores have temperature/accumulation data for at least 150 ka. It would be possible to extend the composite by adding records from e.g. WAIS Divide (67 ka) and Siple Dome (57 ka), but we don't think this would assist with comparison of overall

glacial-interglacial behavior, and – when presented in a composite time series – would misleadingly create the appearance that WAIS data were incorporated for all or most of the 408 ka presented in the figure.

The PC1 lines in Figs. 3j-l correspond to the EOF1 maps in Figs. 3g-i. As noted above, we will add some clarification in the manuscript on how the reader can combine these two pieces of information.

*11. Figure 7: See above comment for Figure 1.*

The color will be updated per the reviewer's suggestion.

*12. Figure 8: Move the legend outside the plot so that it is not overlying the mass balance curves.*

This will be done.

**Reply to Reviewer Comment 2 (Johannes Sutter)**

*1. Methods and ocean forcing.*
*While the method section is clearly written, I think that a more transparent discussion of the strengths and weaknesses of the forcing approach would improve the manuscript and help the reader to put the results into perspective with the literature in the field. The fact that a transient model run spanning several hundred thousand years is used to force the ice sheet model is very impressive, but it should be clearly stated that this is at the expense of resolution which is very coarse. It is for example well known that global ocean models have a hard time resolving circumantarctic circulation which mostly leads to inaccurate representations of warming during interglacials and therefore a muted response of the Antarctic Ice Sheet (e.g. Sutter et al., 2016). I would imagine that the representation of variability of circumantarctic ocean temperatures is even worse in EMICs. I guess this is one of the reasons why ice sheet volume remains relatively high for most interglacials in the manuscript presented here, as well as in Tigchelaar et al. (2018). Throughout the manuscript (Methods, Results, Discussion), it should be re-iterated that with the ocean forcing used in this manuscript, the impact of ocean temperatures on transient interglacial ice sheet dynamics in the late Quaternary cannot be accurately assessed. Upon reading the manuscript, I had the impression that the results shown here imply that ocean temperature forcing in Interglacials or deglaciation phases is not important for ice sheet retreat which would contradict the current literature on how the Antarctic Ice Sheet responds to warmer worlds or in glacial terminations. I am sure that this is not the intended take away message but the chance of misinterpretation for someone not familiar with the field is high.*

Thank you for this comment. We are well aware that the simulated ocean temperature anomalies in LOVECLIM are likely too low, that certain ocean processes (such as Antarctic Bottom Water formation and sub-shelf circulation) are not well captured in EMICs and GCMs, and that the lack of dynamic coupling between ocean and ice sheet precludes inclusion of important feedback mechanisms (e.g., freshwater forcing from meltwater fluxes). It is therefore by no means our intention to have the reader take away that ocean forcing is not important for AIS evolution. We will be more clear about these methodological shortcomings throughout the manuscript to remedy this.

*2. Ice sheet model*
*The description of the ice sheet model is rather compact, which is probably due to the fact, that it has been discussed at length in the cited literature. However, a quick reference as how ice shelf mass balance is treated would be helpful (calving, basal melt parameterization). Providing an assessment of how well the basal ice shelf melt pattern matches the present day observed melt rate (e.g. Depoorter et al., 2013, Rignot et al., 2013) would be helpful as well. How is the model tuned, and how does it perform against present day and paleo benchmarks? Also it would be worth mentioning how the resolution (40 km) used here could affect the results compared to higher resolution studies.*

The ice-sheet model's design involves the use of local parameterizations for important fine-scale processes, that cannot be resolved explicitly by the model grid. In particular, the ice flux across grounding lines is determined by the boundary-layer treatment of Schoof (2007), which captures grounding-line migration well that would otherwise require much finer resolution around the grounding-line zone. Other such features are the sub-grid interpolation of grounding-line location, and fractional areal cover at the edge of ice shelves (Pollard et al., 2012). These features yield model results that are quite independent of horizontal resolution, both in short tests and in long-term runs, for grid sizes between 5 and 40 km (Pollard et al., 2015, Supplemental Information). We will make a mention of this where we describe ice sheet model resolution in Sect. 2.2.

The parameterization of oceanic basal melting under ice shelves uses Eq. 17 in Pollard and DeConto (2012), but has been consolidated with no ad-hoc variations in coefficients (i.e., K=3 everywhere), as described in Pollard et al. (2015, Supplemental Information). As in several other models, the melt rate is proportional (with a single coefficient) to the square of the temperature difference between the base of the ice and the closest grid point at 400 m depth in an ocean dataset. This yields reasonable patterns of modern sub-ice shelf melt, as shown in the Appendix of Pollard et al. (2017), which are generally within ongoing estimates of empirical uncertainties and rapidly changing decadal trends (Depoorter et al., 2013., Rignot et al, 2013). As a design principle, a spatially uniform coefficient is used rather than tuning it point-by-point to match modern estimated maps, because ocean circulation and the associated best-fit coefficients could well change drastically as cavity geometries underneath ice shelves vary over the course of past or future long-term simulations. We will include this equation, parameter values, and a brief discussion of comparison against observations in the manuscript.

As for paleo-benchmarks, previous simulations with this ice sheet model driven by parameterized climates produce realistic expanded grounding line extents and marginal thicknesses at LGM (Mackintosh et al., 2011; Briggs et al., 2013, 2014; Pollard et al., 2016, 2017). The global sea-level fall corresponding to the expanded ice volume is on the low side of the range of estimates (5 to 8 m, Pollard et al., 2016). A greater challenge is achieving good simulations of the last deglacial retreat through time to the present (~20 to 10 ka). This has been a major focus in our and others modeling efforts. We and others have applied large ensembles of model parameter sets, and automated scoring algorithms (Briggs et al., 2013, 2014; Pollard et at 2016, 2017), comparing with several diverse types of paleo data, reviewed for instance in Bentley et al. (2014). This data includes grounding-line positions vs. time, and cosmogenically-derived thinning in inland marginal areas. The general picture is well simulated, including thin streaming ice over much of the major West Antarctic embayments and adjoining ranges (e.g., Stone et al., 2003; Ackert et al., 2007, 2013, Goehring et al., 2019), but with smaller-scale regional disparities (Johnson et al., 2008, 2014, 2017). In general our model captures the LGM state and subsequent deglacial retreat well, to within the general level of uncertainties within the paleo data, and also the modern state of grounded and floating ice (Pollard and DeConto, 2012; Pollard et al., 2016). We will add a brief summary of these comparisons to the Methods section of the paper.

*3. Representation of ocean temperatures*
*The authors mention in section 2.2.2, that LOVECLIM Southern Ocean temperatures are generally too cold. As you use an anomaly forcing to prevent bias propagation it would be interesting how big the glacial and interglacial temperature anomalies (e.g. at 400 m depth) close the ice shelves are.*

As can be seen in Figs. 3i & 3l, the dominant EOF (which explains about 86% of variance) has a glacial-interglacial amplitude of about 0.4 °C. Fig. R1 shows the zonal mean ocean temperature anomaly at 65 °S. It has a glacial-interglacial amplitude of 0.6 °C, with only limited warming during interglacials. To our knowledge there are no near-Antarctic paleo-reconstructions of sub-surface ocean temperatures that could be used to validate this amplitude. We find that simulated Southern Ocean SST variability compares well to e.g., the Ho et al. (2012) record from 54 °S (both an amplitude of ~8°C), but LOVECLIM deep-water temperature anomalies are much lower than those reconstructed from the Elderfield et al. (2012) record. We will add these values to our discussion of simulated ocean temperatures in Sect. 3.1.

[Figure]

**Figure R1** – Zonal mean 400m ocean temperature anomaly at 65 °S

*4. Presentation of Results*
*The presentation and discussion of the results is currently written in a very affirmative manner which sometimes ignores the biases introduced by the experimental setup. For example 3.4.4 suggests that temporal ocean temperature changes are not relevant for ice volume changes. While this is true for the setup used here, it is not the case in multiple publications on the matter (e.g. Golledge et al., 2015, 2017, 2019, DeConto & Pollard 2016, Sutter et al. 2016,2019, Albrecht et al., 2019 TCD). The authors state that they will discuss the validity of the results in the Discussion (p.11 L 10-11), but I have*

*the impression that a serious debate about the shortcomings of the approach and therefore the scope of the results is lacking.*

Based on this comment, we will make sure to be more explicit about methodological shortcomings in Sects. 3.1, 3.4.3, 3.4.4, and the Discussion. Specific responses and suggested edits are given below.

*Specific comments:*

*Title: as the authors focus on the AIS evolution during the last 408 ka I would rename the title to "Nonlinear response of the Antarctic ice sheet to **late-**Quaternary sea level and climate forcing" and use late-Quaternary instead of Quaternary throughout the manuscript (already done in the header of section 3.1).*

This is a good suggestion. We will update the title as suggested, and change occurrences of Quaternary to late-Quaternary in the entire manuscript.

*Check Antarctic Ice Sheet (AIS) throughout the manuscript. Usually it is written in capital initial letters. Furthermore, while you introduce the abbreviation on page 1 L 17 you mostly don't use it later on.*

We will capitalize all instance of Antarctic Ice Sheet, and use the abbreviation AIS where appropriate.

*P2 L10: The importance for what?*

For ice sheet stability. We will add this to this sentence.

*P2 L18-19: I think this is mostly true for glacials but less so for interglacials. While e.g. Konrad et al. (2014) show that a sea level drop due to changes in the gravitational pull during ice loss in interglacials can stabilize the grounding line, the overall rise in sea level during interglacials doesn't play a large role in grounding line retreat as it is mostly limited to just a few meters.*

That is correct, but refers particularly to ice sheet evolution 'beyond' present-day sea level. That is not at odds though with the statement that sea level is an important pacemaker for late-Quaternary ice sheet evolution, because that refers to the entire range of ice sheet configurations in between 'full' glacial and interglacial.

*P2 L22: maybe rephrase to: "particularly leading to a growth of the East Antarctic Ice Sheet (EAIS)"*

This will be changed.

*P3 L11-14: inconsistent use of Section versus Sect.*

This will be changed so that 'Section' if used at the beginning of the sentence, and 'Sect.' elsewhere.

*P4 L1: maybe rephrase to "Each land grid cell ..."*

This will be done.

*P4 L19: rephrase to "While the climate model run closely follows ..., here the longwave radiative effect of CO2 was amplified ..."*

This will be done.

*P4 L34: is the duration of the experiments the reason for the 40km resolution? Then rephrase: Due to limited computational resources and long timescale of the simulations we had to use a relatively coarse resolution of 40 km.*

Yes, the reason for this choice of resolution is mostly that we are running multiple simulations of long duration, but also that – as mentioned above – the results are not substantially different if a resolution of e.g., 20 km is used. We will make the suggested edit.

*P5 L1: rephrase to "Present day climate forcing is obtained from the [...] interpolated to the ice model grid."*

This will be done.

*Do you use the ALBMAP v1 bedrock topography or BEDMAP2 for the initial ice sheet configuration?*
For the initial ice sheet configuration we use the BEDMAP2 bedrock topography, as in Pollard. et al. (2015). We will clarify this in Sect. 2.2.

*P5 L12-30: I would expect the description of the parameterization of the basal shelf melt calculation and calving to be in the ice sheet model section and not in the climate forcing section.*

Because the climate forcing and mass balance calculations are so closely related, it seems more sensible to keep these together. We will rename this section to "Present-day mass balance and climate forcing."

*P6 L6-8: Here I do not understand whether the climate forcing "jumps" every thousand years to a new set of climate anomalies (i.e. the ISM is forced with the same climate anomalies for 1000 years) or whether the transition is smooth. Please clarify.*

It is the former – ice sheet model forced with the same anomalies for 1000 years – though during these 1000 years the climate forcing within the ice sheet model will still evolve in response to simulated changes in elevation. We will clarify this sentence by

changing it to: "The climate forcing in the ice sheet model is updated every 1000 calendar years. Climate anomalies are calculated with respect to the LOVECLIM climatology over the last 200 model years (representing 1000 calendar years) and are bilinearly interpolated to the ice sheet model grid."

*P6 L 8-10: maybe rephrase to:*
*The atmospheric temperature Ta is modified by a lapse rate correction of γ=0.008°Cm-1 to account for surface elevation differences between the reference ice sheet geometry (zobs; Le Brocq et al., 2010) and both the simulated elevation at time t (z(t)), as well as for differences with respect to the LOVECLIM orography (zLC).*

Thank you for the suggestion, we will make this change.

*P7 L1-2: As you force the ISM with ocean temperature anomalies I guess the glacial-interglacial variability is more relevant for the ice sheet's evolution than the present day bias. Please add a sentence which quantifies the ocean warming e.g. in MIS5e and MIS11 and the cooling e.g. during the LGM relative to the PI control climate state (LOVECLIM 1000 year average?).*

Per main comment 3 above, we will add a sentence quantifying glacial-interglacial temperature anomalies to Sect. 3.1, where we discuss late-Quaternary climate evolution. We think this fits better there than in Sect. 2.2.2, which specifically discusses the modeling setup.

*P7 L 11-12: remove sentence "The bottom half of Fig. 2 ..."*

This will be done.

*P7 L26: it would help the reader if CO2 is plotted in Fig. 3j as well to make the pacing more evident.*

Thank you for the suggestion. This will be done.

*P7 L31: Maybe I overlooked this but how do you create the ice core composite? If I understand it correctly you use Dome Fuji, EDC, Vostok, TALDICE and EDML. Only Dome Fuji, EDC, Vostok cover the whole 408 ka.*

The composite was simply constructed as the average of all available ice core temperature/accumulation records for each time in the past, as done in Parrenin et al. (2013), Supplemental Information. We will clarify this in the manuscript.

*P8 L8: Again, you use only one coastal ice core (TALDICE) and 4 interior ice cores. For the latter, the lapse rate correction would be stronger in glacials (e.g. Pollard et al. 2009 and Sutter et al. 2019 TCD Fig. 10). But the biggest discrepancies shown in 3l and 3j occur in Interglacials with too cold ocean and surface temperatures.*

As the two time series are on different scales the discrepancies still exist during glacials, but the reviewer is correct in observing that this does not explain the underestimation of interglacial warming by the climate model. We will change this sentence to "This could partially be due to the fact that the LOVECLIM simulation does not include the lapse rate response to the evolving ice sheet height, but also points at an underestimation of polar temperature change in the climate model, especially during interglacials (Tigchelaar et al. 2018)."

*P8 L17-18: I could imagine that the underestimation of ocean temperature variability in interglacials is the main reason why the ocean forcing is the weakest driver of interglacial ice volume changes in your simulations. This has important implications for your conclusions as this is a methodological bias and not necessarily reflects the actual response of the AIS e.g. in MIS5e and MIS11.*

Per main comment 3 above, we will add a few sentences to this paragraph quantifying the glacial-interglacial temperature anomalies to show these are low compared to reconstructed temperature changes. We will also point out that these modeled anomalies are much lower than the thresholds found to be necessary to simulate ice sheet collapse.

*P8, L23-26: Actually MIS7 shows the lowest surface temperature warming in Antarctic ice cores, how come that for this period the AIS volume is higher than in the other interglacials in your simulations?*

I'm assuming the reviewer means to ask why the AIS volume is lower during this period. As shown in Fig. 3j, MIS7 also has low annual mean surface temperature warming in our climate model simulation. In Tigchelaar et al. (2018) we explained that the reason for the ice sheet loss around 210 ka is the strong summer insolation forcing during this time (Fig. 1a) that drives high summer surface melt and subsequent ice sheet retreat. This actually does not occur at MIS7 (which is a Northern Hemisphere driven termination, Termination III), but during Termination III-a, and illustrates the north-south asynchronicity that can be caused by precessional forcing.

*P9, L12: rephrase to: Figure 5 shows where the individual forcing components have the largest effect on the Antarctic Ice Sheet.*

This will be done.

*P9, L14-15: what is the reason for this thickening? Reduced surface melt? Retreat caused by hydrofracturing?*

As shown in Figs. 4c and 5c, there is a small increase in floating ice volume as well as a small outward expansion of the grounding line during glacial times, so retreat is not the cause for this thickening. Rather, as can be seen in Fig. 8 and is described in Sect. 3.4.2, glacial thickening is primarily caused by a cooling-driven reduction in surface melt and calving rates.

*P9, L16: again I expect this to be caused by the forcing setup and that it is not representative during Interglacials.*

We will add "Due to the small magnitude of the simulated ocean temperature change" to the beginning of this sentence.

*P9, L20: maybe rephrase to: Combined forcing leads to a more pronounced grounding line advance during glacials than in simulations with single forcing.*

We will change this to "Combinations of external forcings lead to a more pronounced grounding line advance during glacials than in simulations with one single forcing."

*P9, L27: rephrase to: Figure 7 depicts the response of grounded ice volume to the respective forcing in the different sensitivity runs.*

This will be done.

*P9, L27: rephrase to: It is important to note that the impact of the sea level forcing in isolation leads to the conversion of grounded into floating ice (during Terminations???).*

We will change this sentence to "As noted before, the impact of sea level forcing in isolation is to convert grounded into floating ice during periods of sea level rise, and the other way around during sea level drops."

*P9, L30: Do surface melt rates really increase in glacials?? The maximum elevation change of ice shelves during glacials would be ca. 120 m.*

Yes, as can be seen in Fig. 8b, surface melt rates increase in this simulation where only sea level is forced to change. The elevation change of 120 m may seem small, but many areas on the perimeter of the AIS have present-day seasonal maximum temperatures that are close to freezing. Only a small increase in annual mean temperature will therefore be needed to generate more Positive Degree Days in the surface melt scheme.

*P10, L5: quantify "fairly consistently".*

We agree this is ambiguous wording and will remove it from this sentence.

*P10, L6: Wording. (Now, ...)*

We will replace 'Now,' with 'In this case'.

*P10, section 3.4.3. This section needs to be expanded, discussing the reasons why the ocean forcing plays a negligible role in the simulations (see main remarks).*

We will add the following sentence to this paragraph: "It is important to note here that this small response to ocean temperature forcing is more likely a function of the low amplitude of the LOVECLIM-simulated ocean temperature forcing than it is indicative of low sensitivity of the AIS to changing ocean conditions, as will be discussed further below."

*P10, L23: rephrase to: Our sensitivity runs show that the simulated response of the AIS to late Quaternary external drivers [...]*

This will be done.

*Section 3.4.4. and the Discussion requires a more detailed disentanglement of what the authors deem to be realistic responses of the AIS to late Quaternary climate and boundary conditions and what they think is due to methodological biases.*

In Sect. 3.4.4. we will change the last sentence to: "Kusahara et al. (2015) also found oceanic melt rates to increase during the Last Glacial Maximum in response to grounding line migration, lending support to these findings. However, as shown in Tigchelaar et al. (2018), the low sensitivity of the modeled AIS to interglacial ocean conditions is likely a result of the low amplitude and resolution of the LOVECLIM ocean temperature forcing and lack of ice-ocean feedbacks in our modeling setup."

In the discussion, we will replace the existing discussion of ocean temperature forcing with the following paragraphs:
"Our finding that ocean temperature forcing plays a limited role in driving changes in Antarctic ice volume contrasts with previous modeling studies of past and future AIS evolution (Golledge et al., 2015; DeConto and Pollard, 2016; Sutter et al., 2016), as well as observations of sustained sub-shelf ice loss in response to ongoing ocean warming at e.g., Pine Island Glacier (Jacobs et al., 2011; Pritchard et al., 2012). This is not surprising, given that the LOVECLIM-simulated ocean temperature anomalies are small (Figs. 3i,l), and ice sheet models typically need ocean warming of 2-5 ºC to initiate interglacial WAIS collapse (Pollard and DeConto, 2009; DeConto and Pollard, 2016; Sutter et al., 2016; Tigchelaar et al., 2018). Absent paleo-reconstructions of near-Antarctic sub-surface ocean temperatures it is difficult to assess how realistic our LOVECLIM simulation is, though critical processes such as Antarctic Bottom Water formation are known to be poorly represented in low-resolution climate models (e.g., Snow et al., 2015), and previous studies have found LOVECLIM in particular to have more muted late-Quaternary temperature variability than other models (Lowry et al., 2019).
In addition, many regional oceanographic processes can affect the circum-Antarctic ocean environment beyond large-scale climate forcing. For example, the blocking effects of sea ice formation (Hellmer et al., 2012), the role of winds in pushing warm waters onto the continental shelf (Thoma et al., 2008; Steig et al., 2012), and the complex geometry of ice shelf cavities (Jacobs et al., 2011; De Rydt et al., 2014) have all been found to be important in observational and modeling studies of current and future oceanic melting of the WAIS ice shelves (Joughin et al., 2014). For that reason,

using 400m-depth Southern Ocean temperatures as the sole driver for sub-shelf melt may miss important near-Antarctic dynamics. Furthermore, melt water fluxes from the AIS have been found to lead to cooling of surface waters and warming at intermediate depth (Menviel et al., 2010; Weber et al., 2014), a feedback mechanism that could increase ice sheet loss (Golledge et al., 2014, 2019). These processes can only really be captured in fully coupled ocean-atmosphere-ice sheet simulations at high resolution, something that is currently not feasible for the long timescales of late-Quaternary climate evolution. However, it should be possible to run shorter simulations – of e.g., the Last Interglacial or Marine Isotope Stage 11 – using such a setup, and perform a similar set of sensitivity experiments as done here. This would likely reveal additional nonlinearities as ice sheet and forcing are allowed to evolve together. The accumulation forcing for instance is similarly impacted by low climate model resolution and lack of ice-climate feedbacks. Time-evolving changes in orography and albedo can substantially alter atmospheric circulation patterns and associated rainfall (Steig et al., 2000; Maris et al., 2014; Steig et al., 2015)."

*P11, L30: I do not understand this sentence. Increased ice loss due to sea level rise induced warming? This needs to more explicit, warming due to sea level driven ice sheet retreat and therefore surface lowering?*

We simply meant to suggest that future studies should not only include greenhouse gas-induced warming, but also include ice sheet retreat due to rising sea levels (from either the Greenland or Antarctic Ice Sheet). To clarify, we will change this sentence to "Further research should therefore explore whether, given the current configuration of grounding line and bedrock, rising sea levels as a result of global warming would further increase or stabilize ice loss.

*P12, L7-8: What is meant by "manually offset"?*

What we meant to suggest here was a series of ice sheet model sensitivity experiments in which the phase relationship between sea level and climate forcing is artificially changed to capture the uncertainties in sea level and greenhouse gas dating. To be more clear in the manuscript, we will change this sentence to "… though future sensitivity runs with the ice sheet model could include artificial shifts in the phase relationship between the sea level and climate forcing to explore associated nonlinearities."

*P12, L10: rephrase, e.g. : In particular, as the ice sheet grows ice sheet areas with higher precipitation expand leading to a positive feedback while at the same time, the ice margin advances into warmer ocean waters which leads to a negative feedback.*

We will change this sentence to: "In particular, as the ice sheet grows the ice sheet expands into areas of higher precipitation, leading to a positive feedback, while at the same time the ice margin advances into warmer ocean waters, causing a negative feedback."

*P12, L15-18: This is the only place in the manuscript which states that LOVECLIM ocean temperature variability is too low and that this could be causal to the muted response during interglacials. Unfortunately, this sentence is right away relativized in the next sentence, citing **one** publication, while a wealth of publications identified ocean warming to be the main driver of ice loss in late Quaternary interglacials (e.g. Golledge et al., 2015,2017,2019, DeConto & Pollard 2016, Sutter et al., 2016).*

As detailed above, we will expand on this discussion in the revised manuscript to better highlight the shortcomings of our methodology. However, it should be noted that in these Golledge, DeConto and Sutter studies, ocean warming is a main driver of ice loss because an ocean forcing is chosen that leads to ice loss, not because high-resolution ocean simulations, circumantarctic temperature reconstructions or a coupled modeling setup dictate a precise level of ocean forcing.

*P12, L28-29: replace sentence "Previous modeling studies have failed to elucidate how these different external drivers interact in driving large glacial ice sheet growth and interglacial sea level highstands." E.g. With "In contrast to previous studies, here we focus on the interaction of different external forcings driving Antarctic Ice Sheet changes". There are previous studies who discuss individual forcing components (e.g. Pollard et al. 2009, de Boer et al. 2013), just not as comprehensive as done here.*

We will make the suggested change.

*P12, L 33-35: I am not fully convinced that this is the case, or at least that this study shows that, as the effect of ocean temperature changes in interglacial ice sheet retreat are not adequately captured in the simulations presented here.*

In the revised manuscript we will change this concluding sentence to further highlight the uncertainty around ocean forcing: "Our modeling setup likely underestimates the role of oceanic forcing, which remains largely unbound by the geologic record and needs to be further explored in a coupled climate-ice sheet modeling framework that can account for critical circumantarctic oceanographic processes."

*Figures 1,4,5,6,8: move labels a),b),c) out of the figures panels.*

This will be done.

[revised manuscript text omitted]